# Detecting adaptive introgression in human evolution using convolutional neural networks

Graham Gower[1]*, Pablo Iáñez Picazo[1], Matteo Fumagalli[2], Fernando Racimo[1]

[1]Lundbeck GeoGenetics Centre, Globe Institute, Faculty of Health and Medical Sciences, University of Copenhagen, Copenhagen, Denmark; [2]Department of Life Sciences, Silwood Park Campus, Imperial College London, London, United Kingdom

**Abstract** Studies in a variety of species have shown evidence for positively selected variants introduced into a population via introgression from another, distantly related population—a process known as adaptive introgression. However, there are few explicit frameworks for jointly modelling introgression and positive selection, in order to detect these variants using genomic sequence data. Here, we develop an approach based on convolutional neural networks (CNNs). CNNs do not require the specification of an analytical model of allele frequency dynamics and have outperformed alternative methods for classification and parameter estimation tasks in various areas of population genetics. Thus, they are potentially well suited to the identification of adaptive introgression. Using simulations, we trained CNNs on genotype matrices derived from genomes sampled from the donor population, the recipient population and a related non-introgressed population, in order to distinguish regions of the genome evolving under adaptive introgression from those evolving neutrally or experiencing selective sweeps. Our CNN architecture exhibits 95% accuracy on simulated data, even when the genomes are unphased, and accuracy decreases only moderately in the presence of heterosis. As a proof of concept, we applied our trained CNNs to human genomic datasets—both phased and unphased—to detect candidates for adaptive introgression that shaped our evolutionary history.

*For correspondence:
graham.gower@gmail.com

**Competing interests:** The authors declare that no competing interests exist.

## Introduction

Ancient DNA studies have shown that human evolution during the Pleistocene was characterised by numerous episodes of interbreeding between distantly related groups (*Green et al., 2010*; *Reich et al., 2010*; *Meyer et al., 2012*; *Prüfer et al., 2017*; *Kuhlwilm et al., 2016*). We now know, for example, that considerable portions of the modern human gene pool derive from Neanderthals and Denisovans (*Green et al., 2010*; *Reich et al., 2010*; *Prüfer et al., 2014*). In the past few years, several methods have been developed to identify regions of present-day or ancient human genomes containing haplotypes that were introgressed from other groups of hominins. These include methods based on probabilistic models (*Sankararaman et al., 2014*; *Steinrücken et al., 2018*; *Racimo et al., 2017a*), on summary statistics (*Vernot and Akey, 2014*; *Vernot et al., 2016*; *Racimo et al., 2017b*; *Durvasula and Sankararaman, 2019*) and on ancestral recombination graph reconstructions (*Kuhlwilm et al., 2016*; *Hubisz et al., 2020*; *Speidel et al., 2019*). Presumably, some of the introgressed material may have had fitness consequences in the recipient populations. While recent evidence suggests that a large proportion of Neanderthal ancestry was likely negatively selected (*Harris and Nielsen, 2016*; *Juric et al., 2016*), there is also support for positive selection on a smaller proportion of the genome—a phenomenon known as adaptive introgression (AI) (*Whitney et al., 2006*; *Hawks and Cochran, 2006*; *Racimo et al., 2015*).

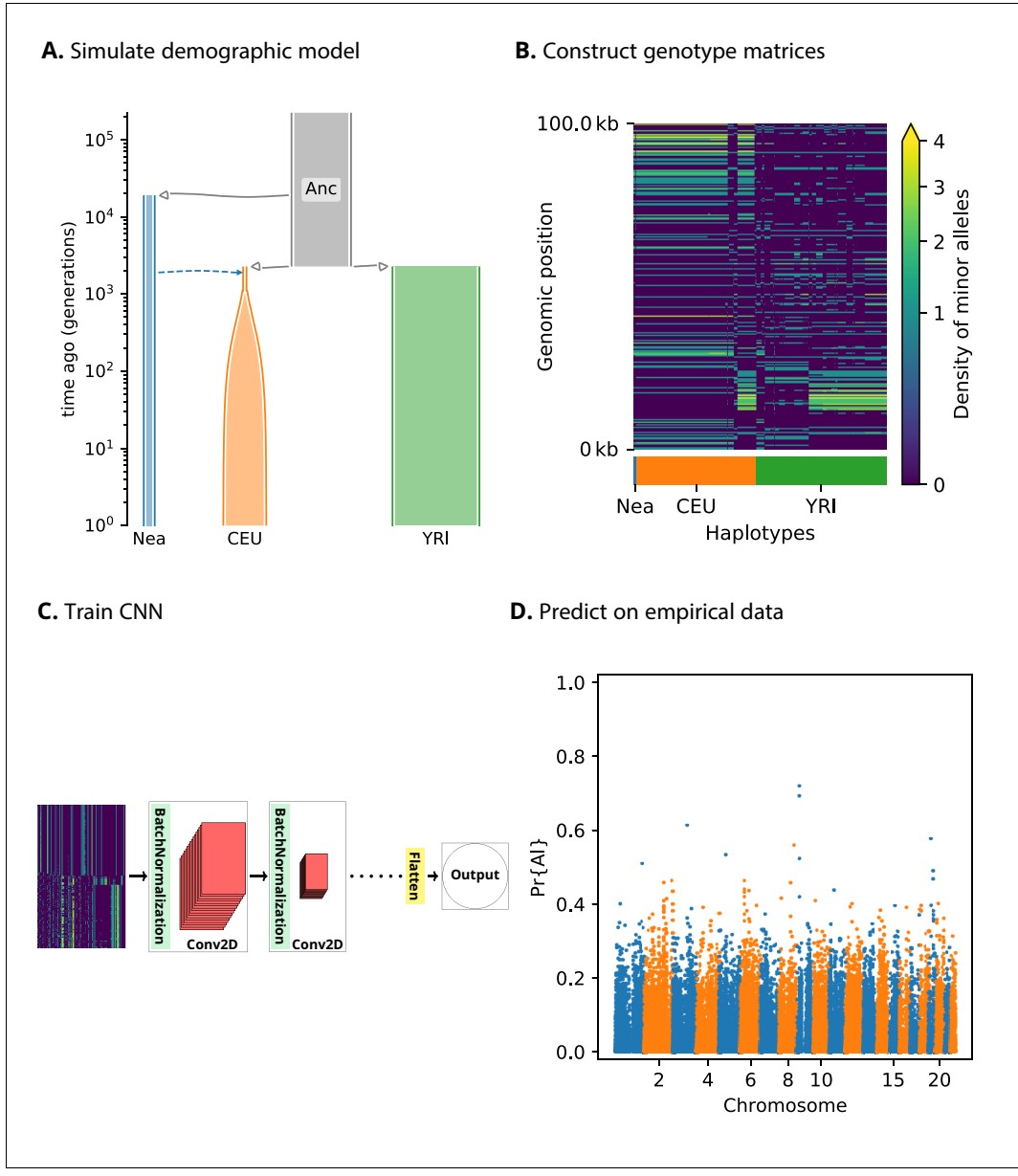

**Figure 1.** A schematic overview of how genomatnn detects adaptive introgression. We first simulate a demographic history that includes introgression, such as Demographic Model A1 shown in (**A**), using the SLiM engine in stdpopsim. Parameter values for this model are given in *Appendix 3—table 1*. Three distinct scenarios are simulated for a given demographic model: neutral mutations only, a sweep in the recipient population, and adaptive introgression. The tree sequence file from each simulation is converted into a genotype matrix for input to the CNN. (**B**) shows a genotype matrix from an adaptive introgression simulation, where lighter pixels indicate a higher density of minor alleles, and haplotypes within each population are sorted left-to-right by similarity to the donor population (Nea). In this example, haplotype diversity is low in the recipient population (CEU), which closely resembles the donor (Nea). Thousands of simulations are produced for each simulation scenario, and their genotype matrices are used to train a binary-classification CNN (**C**). The CNN is trained to output Pr[AI], the probability that the input matrix corresponds to adaptive introgression. Finally, the trained CNN is applied to genotype matrices derived from a VCF/BCF file (**D**).

The online version of this article includes the following figure supplement(s) for figure 1:

**Figure supplement 1.** Schematic overview of Demographic Model A1 and A2.
**Figure supplement 2.** Schematic overview of Demographic Model B.

Genomic evidence for AI has been found in numerous other species, including butterflies (*Pardo-Diaz et al., 2012*; *Enciso-Romero et al., 2017*), mosquitoes (*Norris et al., 2015*), hares (*Jones et al., 2018*), poplars (*Suarez-Gonzalez et al., 2016*), and monkeyflowers (*Hendrick et al., 2016*). A particularly striking example is AI in dogs, which appears to show strong parallels to AI in humans when occupying the same environmental niches. For example, a variant of the gene *EPAS1* has been shown to have introgressed from an archaic human population into the ancestors of Tibetans, and subsequently risen in frequency in the latter population, as a consequence of positive selection to high altitude (*Huerta-Sánchez et al., 2014*). A different high-frequency *EPAS1* variant is also uniquely found in Tibetan Mastiffs, and appears to also have introgressed into this gene pool via admixture with a different species, in this case Tibetan wolves (*Miao et al., 2017*), likely due to the same selective pressures.

To detect AI, researchers can look for regions of the genome with a particularly high frequency of introgressed haplotypes from a donor species or population into a recipient species or population. These haplotypes are often detected assuming neutrality of archaic alleles since the introgression event (*Vernot et al., 2016*; *Vernot and Akey, 2014*; *Sankararaman et al., 2016*). Other studies have designed statistics that are sensitive to characteristic patterns left by AI, using simulations incorporating both admixture and selection (*Gittelman et al., 2016*; *Racimo et al., 2017b*). More recently, *Setter et al., 2020* developed a likelihood framework to look for local alterations to the site frequency spectrum that are consistent with adaptive introgression, using only data from the recipient species. The main challenge that these studies face is that it is hard to jointly model selection from material introduced via admixture (*Racimo et al., 2015*).

To overcome the need to compress data into summary statistics (which might miss important features), deep learning techniques are increasingly becoming a popular solution to address problems in population genetics. These problems include the inference of demographic histories (*Sheehan and Song, 2016*; *Flagel et al., 2019*; *Villanea and Schraiber, 2019*; *Mondal et al., 2019*; *Sanchez et al., 2021*), admixture (*Blischak et al., 2021*), recombination (*Chan et al., 2018*; *Flagel et al., 2019*; *Adrion et al., 2020b*), and natural selection (*Schrider and Kern, 2018*; *Sheehan and Song, 2016*; *Torada et al., 2019*; *Isildak et al., 2021*). Deep learning is a branch of machine learning that relies on algorithms structured as multi-layered networks, which are trained using known relationships between the input data and the desired output. They can be used for classification, prediction, or data compression (*Aggarwal, 2018*). Among the techniques in this field, convolutional neural networks (CNNs) are a family of methods originally designed for image recognition and segmentation (*LeCun and Bengio, 1995*; *Krizhevsky et al., 2012*), which have been recently applied to population genetic data (*Chan et al., 2018*; *Flagel et al., 2019*; *Torada et al., 2019*; *Isildak et al., 2021*; *Blischak et al., 2021*; *Sanchez et al., 2021*). A CNN can learn complex spatial patterns from large datasets that may be informative for classification or prediction, using a series of linear operations known as convolutions, to compress the data into features that are useful for inference.

Despite the recent advances in deep learning for population genetics, no dedicated attempts have been made to identify AI from population genomic data. Here, we develop a deep learning method called genomatnn that jointly models archaic admixture and positive selection, in order to identify regions of the genome under adaptive introgression. We trained a CNN to learn relevant features directly from a genotype matrix at a candidate region, containing data from the donor population, the recipient population and a unadmixed outgroup. The method has >88% precision to detect AI and is effective on both ancient and recently selected introgressed haplotypes. We then applied our method to population genomic datasets where the donor population is either Neanderthals or Denisovans and the recipient populations are Europeans or Melanesians, respectively. In each case, we used the Yoruba population as a unadmixed outgroup and we were able to both recover previously identified AI regions and unveil new candidates for AI in human history.

## Results

### A CNN for detecting adaptive introgression

We assume we have sequence data from multiple populations: the donor population and the recipient population in an admixture event, as well as an unadmixed population that is a sister group to

the recipient (*Figure 1*). Our method relies on partitioning the genomes into windows, which we chose to be 100 kbp in size. For each window, we constructed an $n \times m$ matrix, where $n$ corresponds to the number of haplotypes (or diploid genotypes, for unphased data) and $m$ correspond to a set of equally sized bins along the genomic length of the window. Each matrix entry contains the count of minor alleles in an individual's haplotype (or diploid genotype) in a given bin. Within each population, we sorted these pseudo-haplotypes (or genotypes) according to similarity to the donor population, and concatenated the matrices for each of the populations into a single pseudo-genotype matrix (*Figure 1*).

We designed a CNN (*Figure 1*) that takes this concatenated matrix as input to distinguish between adaptive introgression scenarios and other types of neutral or selection scenarios. The CNN was trained using simulations, and uses a series of convolution layers with successively smaller outputs, to extract increasingly higher level features of the genotype matrices—features which are simultaneously informative of introgression and selection. The CNN outputs the probability that the input matrix comes from a genomic region that underwent adaptive introgression. As our simulations used a wide range of selection coefficients and times of selection onset, the network does not assume these parameters are known a priori, and is able to detect complete or incomplete sweeps at any time after gene flow.

Our method has several innovative features relative to previous population genetic implementations of CNNs (described extensively in the Materials and methods section). For example, when loading the genotype matrices as input, we implemented an image resizing scheme that leads to fast training times, while avoiding the drawbacks of similar methods (*Torada et al., 2019*), by preserving inter-allele distances and thus the local density of segregating sites. Additionally, instead of using pooling layers, we used a $2 \times 2$ step size when performing convolutions. This has the same effect as pooling, in that the output size of the layer is smaller than the input, so the accuracy of the model is comparable to traditional implementations of CNNs, but it has a much lower computational burden (*Springenberg et al., 2015*).

Furthermore, we incorporated a framework to visualise the features of the input data that draw the most attention from the CNN, by plotting saliency maps (*Simonyan et al., 2014*). Saliency maps can help to understand which regions of the genotype matrix contribute the most towards the CNN prediction score.

We also provide downloadable pre-trained CNNs as well as a pipeline for training new CNNs (see Materials and methods). These interface with a new selection module that we designed and incorporated into the stdpopsim framework (*Adrion et al., 2020a*), using the forwards-in-time simulator SLiM (*Haller and Messer, 2019*). We believe this will facilitate the application of the method to other datasets, allowing users to modify its parameters according to the specific requirements of the biological system under study.

## Performance on simulations

We aimed to assess the performance of our method on simulations. We performed simulations under two main demographic models:

- Demographic Model A1: a three-population model including an African, a European and a Neanderthal population, with Neanderthal gene flow into Europeans (*Figure 1* and *Figure 1—figure supplement 1*)
- Demographic Model B: a more complex model, including an African, a Melanesian, a Neanderthal and a Denisovan population, with two pulses of Denisovan gene flow into Melanesians, plus Neanderthal gene flow into non-Africans, based on *Jacobs et al., 2019* (*Figure 1—figure supplement 2*).

When training a CNN on Demographic Model A1 using phased data, we obtained a precision of 90.2% (the proportion of AI predictions that were AI simulations) and 97.9% negative predictive value (NPV; the proportion of 'not-AI' predictions that were either neutral or sweep simulations) (*Figure 2*). The network output higher probabilities for AI simulations with larger selection coefficients, and for older times of onset of selection. We also observed that the network falsely classified neutral simulations as AI more frequently than it falsely classified sweep simulations. When the CNN was trained on this same demographic model assuming genotypes were unphased, the results were very similar, with 88.1% precision and 98.7% NPV.

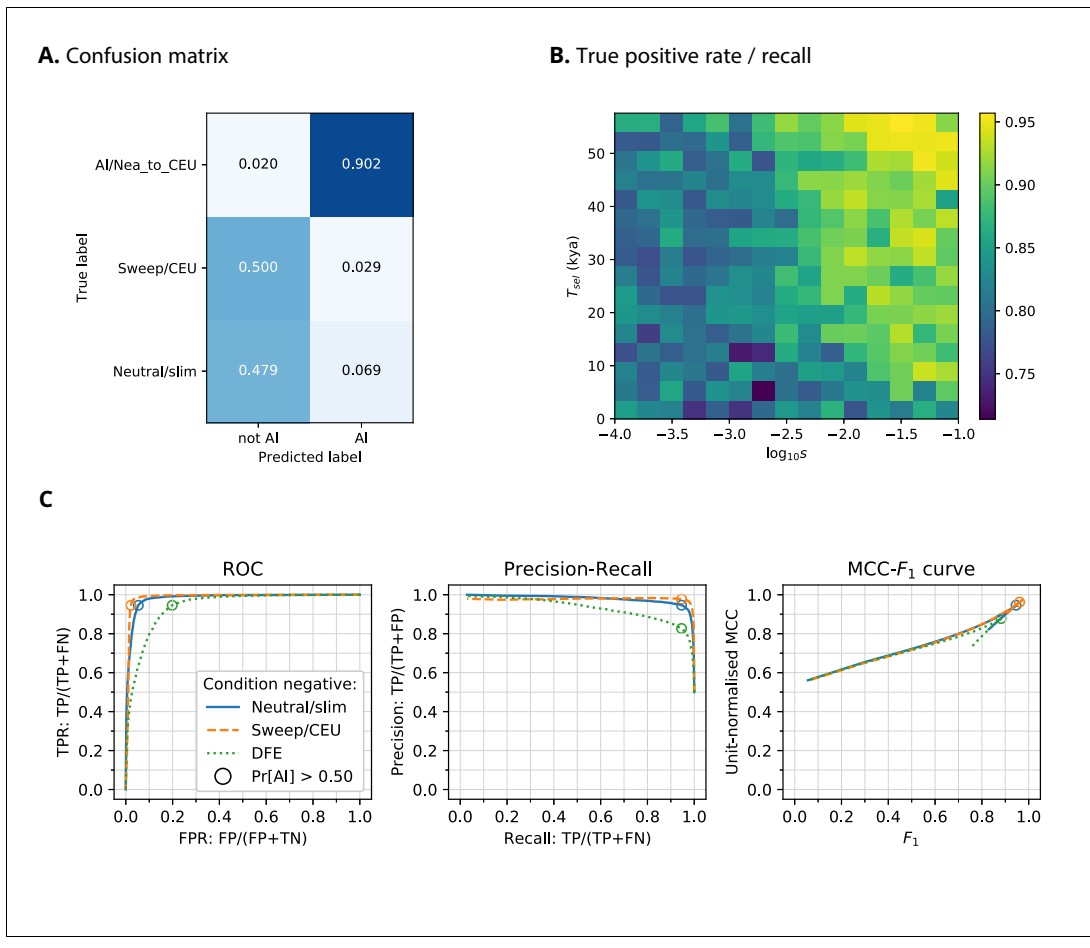

**Figure 2.** CNN performance on validation simulations for Demographic Model A. The CNN was trained using only AI simulations with selected mutation having allele frequency >0.25. (**A**) Confusion matrix. For the two prediction categories, either 'not AI' or AI, we show the proportion attributed to each of the true (simulated) scenarios. (**B**) Average CNN prediction for AI scenarios, binned by selection coefficient, $s$, and time of onset of selection $T_{sel}$. (**C**) ROC curves, precision-recall curves and MCC-$F_1$ curves. The positive condition is AI. The negative conditions are shown using different line styles/colours. The circles indicate the point in ROC-space (respectively Precision-Recall-space, and MCC-$F_1$-space) when using the threshold Pr[AI]>0.5 for classifying a genotype matrix as AI. DFE: distribution of fitness effects. TP: true positives; FP: false positives; TN: true negatives; FN: false negatives; TPR: true positive rate; FPR: false positive rate; ROC: Receiver operating characteristics; MCC: Mathews correlation coefficient; $F_1$: harmonic mean of precision and recall.

The online version of this article includes the following figure supplement(s) for figure 2:

**Figure supplement 1.** Performance evaluation for Demographic Model B.

**Figure supplement 2.** Comparison to other methods and performance evaluation with misspecified demographic models.

When training a CNN on Demographic Model B (assuming unphased genotypes, as accurately phased data are not readily available for Melanesian genomes), we obtained 88.8% precision and 82.5% NPV (*Figure 2—figure supplement 1*). We note here that the network had greater precision when detecting AI derived from the more ancient pulse of Denisovan gene flow than the younger pulse.

*Kim et al., 2018* and *Zhang et al., 2020* recently suggested that introduced genetic material can mask deleterious recessive variation and produce a signal very similar to adaptive introgression. To assess whether heterosis following introgression affects the false positive rates in our CNN, we simulated a distribution of fitness effects (DFE) with recessive dominance for 70% of derived mutations (the rest were simulated as neutral), and found this only slightly increases the false positive rate (*Figure 2*).

## MCC-F$_1$ curve

While precision, recall, and false positive rate are informative, these each consider only two of the four cells in a confusion matrix (true positives, true negatives, false positives, false negatives), and may produce a distorted view of performance with imbalanced datasets (*Chicco, 2017*). To obtain a more robust performance assessment, we plotted the Matthews correlation coefficient (MCC; *Matthews, 1975*) against F$_1$-score (the harmonic mean of precision and recall) for false-positive-rate thresholds from 0 to 100 (*Figure 2*, *Figure 2—figure supplement 1*, *Figure 2—figure supplement 2*), as recently suggested by *Cao et al., 2020*. MCC produces low scores unless a classifier has good performance in all four confusion matrix cells, and also accounts for class imbalance. In MCC-F$_1$ space, the point (1, 1) indicates perfect predictions, and values of 0.5 for the (unit-normalised) MCC indicate random guessing. These results confirm our earlier findings, that the CNN performance is excellent for Demographic Model A1 when considering either neutral and sweep simulations as the condition negative, and performance decreases slightly when DFE simulations are the negative condition (*Figure 2*). Furthermore, the CNN performance is not as good for Demographic Model B, but this is unlikely to be caused by using unphased genotypes (*Figure 2—figure supplement 1* and *Figure 2—figure supplement 2*).

## Comparison to other methods

We compared the performance of our CNN to VolcanoFinder (*Setter et al., 2020*), which scans genomes of the recipient population for patterns of diversity indicative of AI using a coalescent-based model of adaptive introgression (*Figure 2—figure supplement 2*). However, this method only incorporates information from a single population and is designed to detect 'ghost' introgression in cases where the source is not available. VolcanoFinder performed poorly for the demographic models considered here—in some cases, worse than guessing randomly. We also compared our CNN to an outlier-based approach for a range of summary statistics that are sensitive to AI (*Racimo et al., 2017b*). Our CNN is closest to a perfect MCC-F$_1$ score for Demographic model A1 and B, closely followed by the Q95(1%, 100%) and then U(1%, 20%, 100%) statistics developed in *Racimo et al., 2017b*.

## Demographic model misspecification

We then tested robustness to demographic misspecification, by evaluating the CNN trained on Demographic Model A1 against simulations for two other demographic models (*Figure 2—figure supplement 2*). We considered weak misspecification, where the true demographic history is similar to Demographic Model A1 but also includes archaic admixture within Africa following *Ragsdale and Gravel, 2019* (Demographic Model A2; *Figure 1—figure supplement 1*). This resulted in only a small performance reduction. We also considered strong misspecification, where the true demographic history is Demographic Model B. As there are more Melanesian individuals than European individuals in our simulations (because we aimed to mimic the real number of genomes available in our data analysis below), we downsampled the Melanesian genomes to match the number of European genomes, so as to perform a fair misspecification comparison. In this case, the performance of the CNN was noticeably worse than that of the summary statistics, but still better than VolcanoFinder. We note that the summary statistics performance decreased also, to match their performance for the correctly-specified assessments on Demographic Model B. Interestingly, we found that the Q95(1%, 100%) statistic was the most robust method for both cases of misspecification.

## Network attention

To understand which features of the input matrices were used by the CNN to make its predictions, we constructed saliency maps (*Simonyan et al., 2014*). This technique works by computing the gradient of a network's output with respect to a single input. Thus, highlighted regions from the saliency map indicate where small changes in the input matrix have a relatively large influence over the CNN output prediction.

We calculated an average saliency map for each simulation scenario (neutral, sweep, or AI), for a CNN trained on Demographic Model A1 (*Figure 3*). Our results show that when the network was presented with an AI matrix, it focused most of the attention on the Neanderthal and European haplotypes, rather than on the African haplotypes. In non-AI scenarios, the network focused sharply on

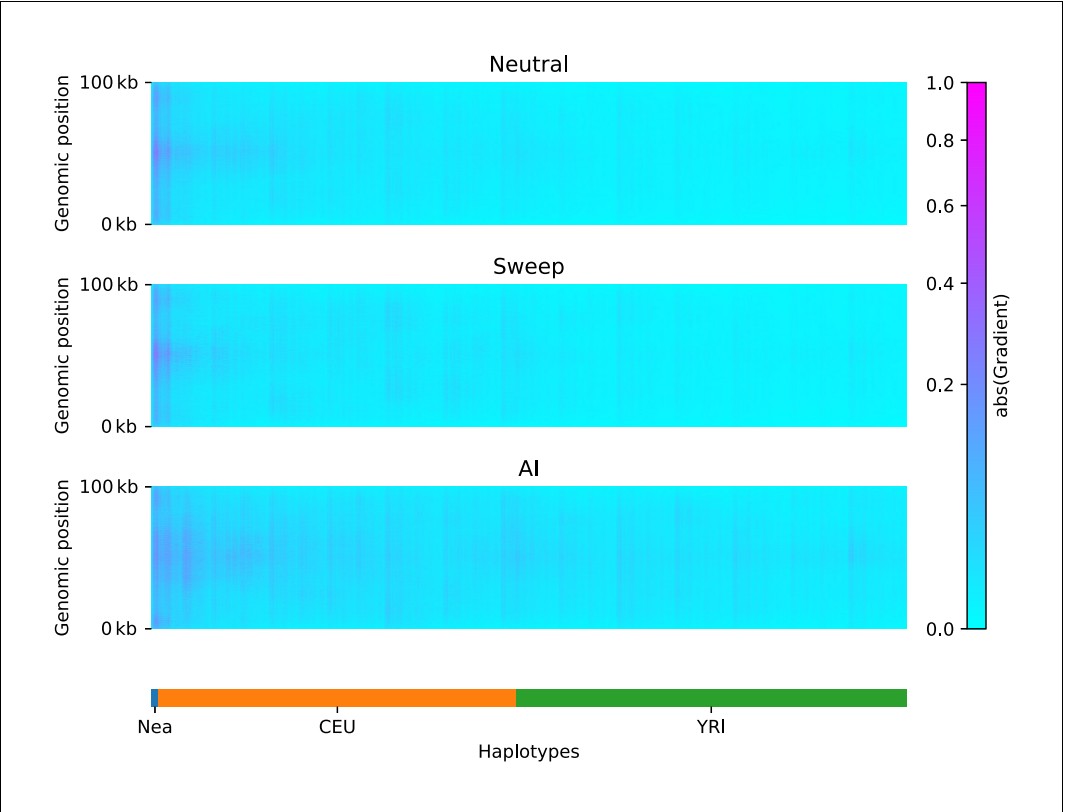

**Figure 3.** Saliency maps, showing the CNN's attention across the input matrices for each simulated scenario, calculated for the CNN trained on Demographic Model A, filtered for beneficial allele frequency >0.25. Each panel shows the average gradient over 300 input matrices encoding either neutral (top), sweep (middle), or AI (bottom) simulations. Pink/purple colours indicate larger gradients, where small changes in the genotype matrix have a relatively larger influence over the CNN's prediction. Columns in the input matrix correspond to haplotypes from the populations labelled at the bottom.

the Neanderthal and left-most European haplotypes. The saliency maps also show a concentration of attention in the central region of the genomic window, closer to where the selected mutation was drawn (even though this mutation does not exist in neutral simulations, and was removed from sweep and AI simulations before constructing genotype matrices; see Methods). We also note a periodic vertical banding pattern in the saliency maps, corresponding to the filter width for the convolution layers.

## Calibration

We implemented a score calibration scheme to account for the fact that our simulation categories (neutrality, sweep, and AI) will be highly imbalanced in real data applications (*Guo et al., 2017*; *Kull et al., 2017*). CNN classifiers sometimes produce improperly calibrated probabilities (*Guo et al., 2017*). In our case, this occurs because the proportion of each category is not known in reality, and thus does not match the simulated proportion. For this reason, we fitted our calibration procedure using training data resampled with various ratios of neutral:sweep:AI simulations (*Figure 4*). We tested different calibration methods by fitting the calibrator to the training dataset, and inspecting reliability plots and the sum of residuals on a validation dataset (see Materials and methods, *Figure 4—figure supplement 1*, *Figure 4—figure supplement 2*, *Figure 4—figure supplement 3*, *Figure 4—figure supplement 4*). When the probabilities are calibrated for even class ratios, Manhattan plots show a large number of high probability candidates across the genome, which obscure the strongest peaks (*Figure 4*). However, once calibrated for class ratios that are skewed towards the neutral class, strong candidates for AI become more apparent.

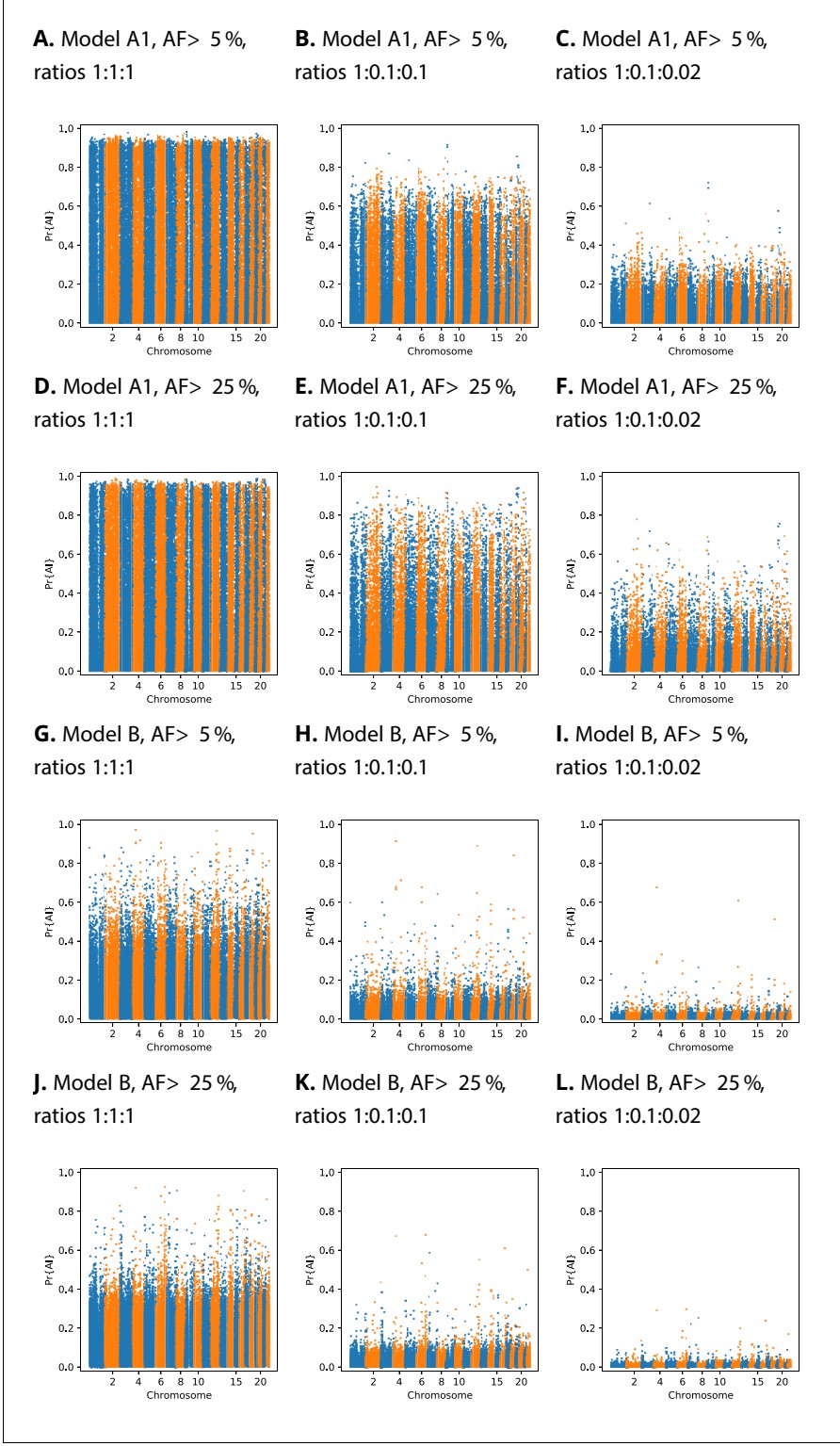

**Figure 4.** Comparison of Manhattan plots using beta-calibrated output probabilities for different class ratios. Each row indicates a single CNN, with equivalent data filtering. Each column indicates different class ratios used for calibration (Neutral:Sweep:AI). AF = Minimum beneficial allele frequency.

The online version of this article includes the following figure supplement(s) for figure 4:

**Figure supplement 1.** Reliability plots for Demographic Model A1 with AF > 5%.

**Figure supplement 2.** Reliability plots for Demographic Model A1 with AF > 25%.

*Figure 4 continued on next page*

**Figure supplement 3.** Reliability plots for Demographic Model B with AF > 5%.
**Figure supplement 4.** Reliability plots for Demographic Model B with AF > 25%.

## Candidates for Neanderthal adaptive introgression in European genomes

We applied our method to a combined genomic panel of archaic hominins (*Prüfer et al., 2017*; *Meyer et al., 2012*) and present-day humans (The 1000 *Auton et al., 2015*; *Jacobs et al., 2019*), to look for regions of the genome where Non-African humans show signatures of AI from archaic hominins. First, we looked for Neanderthal introgression into the ancestors of Northwestern Europeans (CEU panel), using Yoruba Africans (YRI panel) as the unadmixed sister population. To give the network the best chance of avoiding false positives, we tried two different beneficial-allele frequency cutoffs for training: 5% and 25% (*Table 1* and *Appendix 1—table 1*).

We focus here on describing the results from the 25% condition (*Figure 5* and Appendix 4). We found several candidate genes for AI that have been reported before (*Sankararaman et al., 2014*; *Vernot and Akey, 2014*; *Gittelman et al., 2016*; *Racimo et al., 2017b*), including *BNC2*, *KCNQ2/EEF1A2 WRD88/GPATCH1* and *TANC1*. Notably, the candidate region we identify on chromosome 2 around *TANC1* extends farther downstream of this gene, also overlapping *BAZ2B* (*Appendix 4—figure 3*). This codes for a protein related to chromatin remodelling, and may have a role in transcriptional activation. Mutations in *BAZ2B* have recently been associated with neurodevelopmental disorders, including developmental delay, autism spectrum disorder and intellectual disability (*Scott et al., 2020*).

Additionally, we found two novel candidates for AI that have not been previously reported, spanning the regions chr6:28.18 Mb–28.32 Mb (*Appendix 4—figure 7*) and chr20:62.1 Mb–62.28 Mb (*Appendix 4—figure 13*), including multiple genes encoding zinc finger proteins. UK-biobank PheWAS associations (*Canela-Xandri et al., 2018*) suggest both regions generally affect phenotypes

**Table 1.** Top ranking gene candidates corresponding to Neanderthal AI in Europeans.
We show genes which overlap, or are within 100 kbp of, the 30 highest ranked 100 kbp intervals. Adjacent intervals have been merged. The CNN was trained using only AI simulations with selected mutation having allele frequency >0.25, and subsequently calibrated with resampled neutral:sweep:AI ratios of 1:0.1:0.02.

| Chrom | Start | End | Genes |
|---|---|---|---|
| 1 | 104500001 | 104600000 | |
| 2 | 109360001 | 109460000 | LIMS1; RANBP2; CCDC138; EDAR |
| 2 | 160160001 | 160280000 | TANC1; WDSUB1; BAZ2B |
| 3 | 114480001 | 114620000 | ZBTB20 |
| 4 | 54240001 | 54340000 | SCFD2; FIP1L1; LNX1 |
| 5 | 39220001 | 39320000 | FYB; C9; DAB2 |
| 6 | 28180001 | 28320000 | ZSCAN16-AS1; ZSCAN16; ZKSCAN8; ZSCAN9; ZKSCAN4; NKAPL; PGBD1; ZSCAN31; ZKSCAN3; ZSCAN12; ZSCAN23 |
| 8 | 143440001 | 143560000 | TSNARE1; BAI1 |
| 9 | 16700001 | 16820000 | BNC2 |
| 12 | 85780001 | 85880000 | ALX1 |
| 19 | 20220001 | 20380000 | ZNF682; ZNF90; ZNF486 |
| 19 | 33580001 | 33740000 | RHPN2; GPATCH1; WDR88; LRP3; SLC7A10 |
| 20 | 62100001 | 62280000 | CHRNA4; KCNQ2; EEF1A2; PPDPF; PTK6; SRMS; C20orf195; HELZ2; GMEB2; STMN3; RTEL1; TNFRSF6B; ARFRP1; ZGPAT; LIME1; SLC2A4RG; ZBTB46 |
| 21 | 25840001 | 25940000 | |

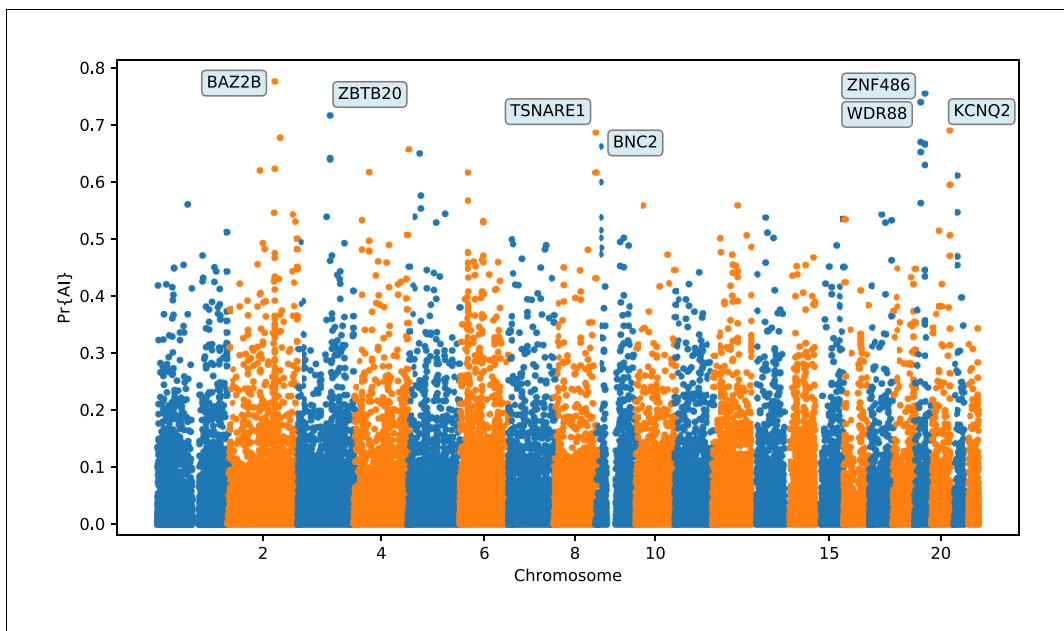

**Figure 5.** Application of the trained CNN to the Vindija and Altai Neanderthals, and 1000 genomes populations YRI and CEU. The CNN was applied to overlapping 100 kbp windows, moving along the chromosome in steps of size 20 kbp. The CNN was trained using only AI simulations with selected mutation having allele frequency > 25%, and subsequently calibrated with resampled neutral:sweep:AI ratios of 1:0.1:0.02.

related to blood, including platelet, erythrocyte and leukocyte counts (at the $p<10^{-8}$ association level, the chr6 region has 91 hits, while the chr20 region has 19, with 10 of these traits in common).

## Candidates for Denisovan adaptive introgression in Melanesian genomes

We then looked for Denisovan AI in Melanesian genomes from the IGDP panel (*Jacobs et al., 2019*), also considering Yoruba Africans as the unadmixed sister group, and using two different beneficial-allele frequency cutoffs for training: 5% and 25% (*Table 2* and *Appendix 1—table 2*).

Again, we focus on describing the results from the 25% condition (*Figure 6* and Appendix 5) Among the top candidates, we found a previously reported candidate for AI in Melanesians: *TNFAIP3* (*Vernot et al., 2016*; *Gittelman et al., 2016*). Denisovan substitutions carried by the introgressed haplotype in this gene have been found to enhance the immune response by tuning the phosphorylation of the encoded A20 protein, which is an immune response inhibitor (*Zammit et al., 2019*).

We found evidence for Denisovan AI in Melanesians at several other candidate regions. A few of these regions (or contiguous regions) were previously reported by *Sankararaman et al., 2016* but not extensively described, possibly because the previously reported sections of those regions deemed to be introgressed were intergenic. One of the regions with strong evidence for AI (chr7:25.1 Mb–25.2 Mb; *Appendix 5—figure 8*) overlaps the *CYCS* gene. This gene codes for cytochrome C: a small heme protein that plays a crucial role in the electron transport chain in mitochondria, and has been associated with various blood-related diseases, like thrombocytopenia (*Morison et al., 2008*; *De Rocco et al., 2014*; *Uchiyama et al., 2018*). Another top candidate region (chr12:108.24–108.34 Mb, *Appendix 5—figure 13*) is upstream of *PRDM4* and *ASCL4*. The former gene codes for a transcription factor that may be involved in the nerve growth factor cell survival pathway and might play a role in tumour suppression (*Yang and Huang, 1999*). The latter gene codes for a different transcription factor that may be involved in skin development (*Jonsson et al., 2004*).

We detected signatures of Denisovan AI in a region in chromosome 3 near *SUMF1* and *LRNN1* (*Appendix 5—figure 2*), which was also identified in *Jacobs et al., 2019*. *SUMF* codes for an

**Table 2.** Top ranking gene candidates corresponding to Denisovan AI in Melanesians.
We show genes which overlap, or are within 100 kbp of, the 30 highest ranked 100 kbp intervals. Adjacent intervals have been merged. The CNN was trained using only AI simulations with selected mutation having allele frequency >0.25, and subsequently calibrated with resampled neutral:sweep:AI ratios of 1:0.1:0.02.

| Chrom | Start | End | Genes |
|---|---|---|---|
| 2 | 129960001 | 130060000 | |
| 3 | 3740001 | 3840000 | SUMF1; LRRN1 |
| 4 | 41980001 | 42080000 | TMEM33; DCAF4L1; SLC30A9; BEND4 |
| 5 | 420001 | 520000 | PDCD6; AHRR; C5orf55; EXOC3; CTD-2228K2.5; SLC9A3; CEP72 |
| 6 | 74640001 | 74740000 | |
| 6 | 81960001 | 82060000 | |
| 6 | 137920001 | 138120000 | TNFAIP3 |
| 7 | 25100001 | 25200000 | OSBPL3; CYCS; C7orf31; NPVF |
| 7 | 38020001 | 38120000 | EPDR1; NME8; SFRP4; STARD3NL |
| 7 | 121160001 | 121260000 | |
| 8 | 3040001 | 3140000 | CSMD1 |
| 12 | 84640001 | 84740000 | |
| 12 | 108240001 | 108340000 | PRDM4; ASCL4 |
| 12 | 114020001 | 114280000 | RBM19 |
| 14 | 61860001 | 61960000 | PRKCH |
| 14 | 63120001 | 63220000 | KCNH5 |
| 14 | 96700001 | 96820000 | BDKRB2; BDKRB1; ATG2B; GSKIP; AK7 |
| 15 | 55260001 | 55400000 | RSL24D1; RAB27A |
| 16 | 62600001 | 62700000 | |
| 16 | 78360001 | 78460000 | WWOX |
| 18 | 22060001 | 22160000 | OSBPL1A; IMPACT; HRH4 |
| 22 | 19040001 | 19140000 | DGCR5; DGCR2; DGCR14; TSSK2; GSC2; SLC25A1; CLTCL1 |

enzyme involved in the hydrolysis of sulfate esters, which has been associated with sulfatase deficiency (*Cosma et al., 2003*). *LRNN1* encodes a protein involved in neuronal differentiation, which has been associated with neuroblastoma and Alzheimer's disease (*Bai et al., 2014*; *Hossain et al., 2012*). Another candidate region is in chromosome 7 and is upstream of *SFRP4* (*Appendix 5—figure 9*), which encodes a protein associated with diabetes (*Mahdi et al., 2012*) and Pyle's disease (*Kiper et al., 2016*). Moreover, there is also a candidate region upstream of *RAB27A*, in chromosome 15 (*Appendix 5—figure 18*). Mutations in this gene cause Griscelli syndrome, which results in pigmentary dilution in the hair and skin, as well as melanosome accumulation in melanocytes (*Ménasché et al., 2000*). Finally, we found evidence for Denisovan AI in two nearby regions in chromosome 14 (*Appendix 5—figure 15* and *Appendix 5—figure 16*). One of these overlaps with *PRKCH*—encoding a protein kinase associated with cerebral infarction (*Kubo et al., 2007*). The other overlaps with *KCNH5*—coding for a potassium channel that may be associated with epileptic encephalopathy (*Veeramah et al., 2013*).

## Discussion

We have developed a new method to detect adaptive introgression along the genome using convolutional neural networks. The method has high precision when reporting candidate AI loci, and high negative predictive value when rejecting loci as not-AI: we obtain greater than 90% accuracy under a variety of different selection scenarios (*Appendix 2—table 1*), with low false positive rates.

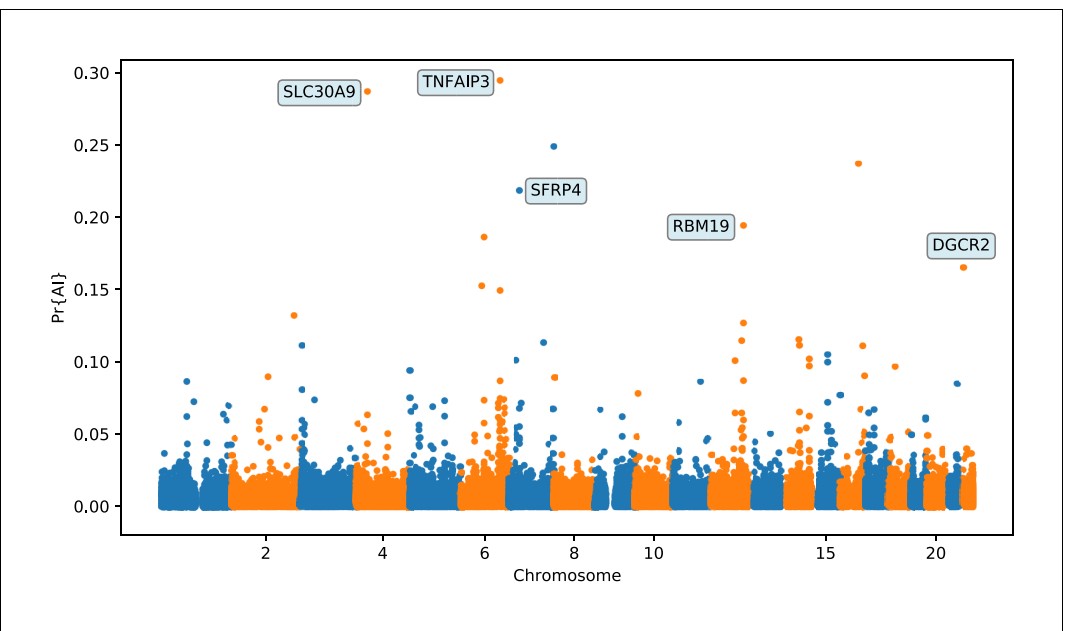

**Figure 6.** Application of the trained CNN to the Altai Denisovan and Altai Neanderthal, 1000 genomes YRI populations, and IGDP Melanesians. The CNN was applied to overlapping 100 kbp windows, moving along the chromosome in steps of size 20 kbp. The CNN was trained using only AI simulations with selected mutation having allele frequency > 25%, and subsequently calibrated with resampled neutral:sweep:AI ratios of 1:0.1:0.02.

As reported previously (*Kim et al., 2018*; *Zhang et al., 2020*), heterosis following introgression can produce patterns very similar to AI, and we found this can inflate false positive detection of AI by our CNN to a small extent. However, we simulated a DFE with recessive dominance for all mutations, which is not realistic in general, so our results in this regard represent a worst-case scenario. A possible future improvement would be to train the CNN on simulations incorporating heterosis. We did not attempt this here because realistic DFE simulations represent a substantial computational burden.

When the demographic model is correctly specified, our CNN performed better than using summary statistics, although the Q95(1%, 100%) statistic (*Racimo et al., 2017b*) also performed remarkably well. This statistic captures high-frequency derived alleles that are shared between the donor and recipient population, to the exclusion of a non-introgressed sister population—intuitively, these are the same features we expect our CNN to be learning. Because of its relative robustness to model misspecification, an outlier approach based on Q95 may be a better choice than our CNN when there is uncertainty regarding the demographic history of the study system. We also found that VolcanoFinder performed very poorly across all our tests, but this is arguably an unfair comparison because it only incorporates information from a single population, and *Setter et al., 2020* themselves found that their method has low sensitivity when the donor population split from the recipient population recently (on the order of $N$ generations ago for Neanderthals/Denisovans and humans).

The CNN took approximately 15 min to train on one NVIDIA Tesla T4 GPU, which amounts to 60 CPU hours for an equivalent CPU-only training procedure. All data were loaded into memory, which required approximately 120 GB RAM during training. The computational bottleneck lay in the generation of SLiM forward simulations: 300,000 simulations took approximately 80 weeks of CPU time for each of demographic models A1 and B. In the future, considerable speedups could potentially be obtained by optimising the simulation step, perhaps by implementing an adaptive introgression simulation framework that takes advantage of the backwards coalescent (e.g. building on the work by *Setter et al., 2020*).

We applied the method to human data, to look for adaptive introgression from archaic humans into the ancestors of present-day human genomes. When looking for Neanderthal AI in European genomes, we found previously reported candidate genes (*BNC2, WRD88/GPATCH1, KCNQ2/ EEF1A2, TANC1/BAZ2B*). We also recovered candidates for adaptive introgression from Denisovans

by applying our method to unphased Melanesian genomes. The top candidates include *TNFAIP3*, which has been reported before, but also include other, novel regions, containing genes involved in blood diseases (*CYCS*), neurological diseases (*PRKCH, KCNH5, LRNN1*), metabolism (*SFRP4, SUMF1*), and skin development (*ASCL4, RAB27A*).

We note, however, that, as with previous methods, visual inspection of the haplotypes or genotypes of the top candidate regions remains a necessary criterion to accurately assess whether a region may have been under adaptive introgression. For example, in the scans we performed, we found a few candidate regions for Neanderthal AI in Europeans that are likely to be false positives, for example chr2:109360001–109460000 (*Appendix 4—figure 2*); chr4:54240001–54340000 (*Appendix 4—figure 5*); chr8:143440001–143560000 (*Appendix 4—figure 8*). These appear to be the result of shared ancestral variation between European and African populations, and yet are classified as having high probability of being under AI. These regions also appear to be generally low in diversity, which is possibly a result of data missingness rather than low diversity per se. Thus, our method allows for a rapid scan and prioritisation of potential targets, but these need to be further assessed with care. Inclusion of more complex selection scenarios, involving positive or balancing selection on ancestral variation, as well as linked selection, might serve to ameliorate the rate of false positives in the future. Furthermore, our simulation procedure does not model genotype errors or variation in data missingness. Not explicitly accounting for this may negatively impact the robustness of the minor allele density computation and the subsequent haplotype sorting procedure, and, in turn, affect the accuracy of the CNN.

Conversely, there may be regions under AI that are classified as highly probable by the CNN, but that did not appear in our top candidates. Validating a large number of candidates might be difficult, but one could imagine running a differently trained CNN (perhaps one better tailored to distinguish AI from more similar scenarios, like selection on shared ancestral variation) on the subset of the regions that are predicted to be AI using a lenient probability cutoff. One could also use our method more generally, to assess the impact of AI across the genome, by comparing the distribution of probability scores with those of simulation scenarios under different amounts of admixture and selection, though in that case one would need to train the CNN on a wider range of admixture rates and demographic models.

The performance of our method necessarily depends upon the demographic history of the populations involved. We found it more challenging to detect AI when the timing of gene flow is younger or the introgressing population is more diverged from the panel that is used to represent it. This is apparent when comparing results for the Neanderthal-into-European demographic scenario and the Denisovan-into-Melanesian demographic scenario. In the former, gene flow is older (~55 kya versus ~50 kya and ~30 kya) (*Sankararaman et al., 2016*; *Jacobs et al., 2019*) and sequences are available for a population closely related to the putative source, which increases power. Furthermore, for the two putative pulses of Denisovan gene flow (*Jacobs et al., 2019*), we find our model has greater recall with AI for the more ancient pulse (94% versus 83.6%; *Figure 2—figure supplement 1*), likely because haplotypes from the older pulse have more time to rise in frequency. Similarly, recall is diminished when the onset of selection is more recent. We also found that distinguishing AI from a selective sweep (hard or soft), is relatively easier than distinguishing AI from neutral variation.

Our method requires sequencing data from the population from which the introgression event originated. This may be problematic in cases where the source of introgression may be distantly related to the population genomic panel that is used to represent it. Future work could involve developing a CNN that can detect adaptive introgression from a ghost (unsampled) population, for cases in which genomic data from the source is unavailable (e.g. see *Setter et al., 2020*).

The method can take either phased or unphased data as input. This flexibility allows for its application to a range of study systems in the future, in which phasing may not be financially or methodologically feasible. It does, however, require called genotypes and is therefore not yet suitable for genomes sequenced at low coverage. One could envision extending the framework developed here to low-coverage genomes by working with matrices of genotype likelihoods (*Korneliussen et al., 2014*) rather than matrices of genotypes or haplotypes. *Flagel et al., 2019*, for example, developed a CNN to infer recombination rates in tetraploids without genotype calls, using read pileup information. A CNN could learn the relationship between observed read counts or genotype likelihoods under a given adaptive introgression scenario and the model parameters that can generate that data, but we leave that to a future work.

Future studies could also address the fact that we must use simulations to train the network, which involves an implicit amount of supervision by the user. The range of parameters and models that are simulated during training are necessarily specified a priori, and misspecification can negatively affect CNN performance. Progress in this regard could involve the use of generative adversarial networks (GANs), which appears to be a fruitful way to address this. Indeed, recent work suggests that one can train a GAN to learn to generate realistic population genomic data for any population (*Wang et al., 2020*).

The attention analyses performed here allowed only a posteriori reasoning on how the network learned to predict AI, so further work is needed in this area. For instance, interpretability of neural networks can be assessed using symbolic metamodelling (*Alaa and van der Schaar, 2019*) with reinforcement learning algorithms deployed to identify the subset of most informative features of input data (*Yoon et al., 2019*). In this context, such approaches should be able to pinpoint the important characteristics of genomic data, and possibly derive more informative summary statistics to predict complex evolutionary events.

In summary, we have shown that CNNs are a powerful approach to detecting adaptive introgression and can recover both known and novel selection candidates that were introduced via admixture. As in previous applications to other problems in the field (*Sheehan and Song, 2016*; *Flagel et al., 2019*; *Schrider and Kern, 2018*; *Villanea and Schraiber, 2019*; *Mondal et al., 2019*; *Torada et al., 2019*; *Isildak et al., 2021*), this exemplifies how deep learning can serve as a very powerful tool for population genetic inference. This type of technique may thus be a useful resource for future studies aiming to unravel our past history and that of other species, as statistical methodologies and computational resources continue to improve.

## Materials and methods

### Simulations

For CNN training, we performed simulations under three scenarios: neutral mutations only; positive selection of a de novo mutation in the recipient population (selective sweep); and positive selection of a derived mutation that was transferred via gene flow from the donor population to the recipient population (adaptive introgression, AI). In the sweep and AI scenarios, the selection coefficient was drawn log-uniformly from between 0.0001 and 0.1 for Europeans and between 0.001 and 0.1 for Melanesians (in the latter case, very few selected alleles survive with a very small selection coefficient, so we narrowed the range to reduce computational burden). The uniformly distributed time of mutation was decoupled from the uniformly distributed time of selection onset, thus allowing for soft sweeps (*Hermisson and Pennings, 2005*). For the selective sweep scenario, the mutation and selection times could occur at any time older than 1 kya but more recent than the split between the recipient population and its unadmixed sister population, with the constraint that the mutation must be introduced before the onset of selection. For the AI scenario, a neutrally evolving mutation was introduced to the donor population any time more recent than the split between the donor and the ancestor of recipient and unadmixed sister population, but older than 1 kya before the introgression event. Then, this mutation was transmitted to the recipient population, whereupon selection could start to act on it at any time after introgression but before 1 kya.

We further evaluated our Demographic Model A1 CNNs using an additional 10,000 simulations that incorporated a DFE using the parameters estimated for Europeans in *Kim et al., 2017* and used in *Kim et al., 2018*. We considered two mutation types: 30% neutral and 70% deleterious. The deleterious portion of introduced mutations had a selection coefficient drawn from a reflected gamma distribution with shape parameter 0.186, and expected value $-0.01314833$. We approximated the dominance scheme from *Kim et al., 2018*, using a fixed dominance coefficient for deleterious mutations of $0.5/(1 - 7071.07 * E[s])$ where $E[s]$ is the expected value from the gamma distribution (i.e. all deleterious mutations were effectively recessive).

To incorporate selection, we implemented a new module in stdpopsim (*Adrion et al., 2020a*), which leverages the forwards-in-time simulator SLiM (*Haller and Messer, 2019*) for simulating selection. For consistency, we also used stdpopsim's SLiM engine for neutral simulations. stdpopsim uses SLiM's ability to output tree sequences (*Haller et al., 2019*; *Kelleher et al., 2018*), which retains complete information about the samples' marginal genealogies. Further, stdpopsim recapitates the

tree sequences (ensuring that all sampled lineages have a single common ancestor), and applies neutrally evolving mutations to the genealogies, using the coalescent framework of msprime (*Kelleher et al., 2016*).

We simulated 100 kbp regions, with a mutation rate of $1.29 \times 10^{-8}$ per site per generation (*Tian et al., 2019*), an empirical recombination map drawn uniformly at random from the HapMapII genetic map (*Frazer et al., 2007*), and the selected mutation introduced at the region's midpoint. For both the sweep scenario and the AI scenario, we used a rejection-sampling approach to condition on the selected allele's frequency being $\geq$ in the recipient population at the end of the simulation. This was done by saving the simulation state prior to the introduction of the selected mutation (and saving again after successful transmission to the recipient population, for the AI scenario), then restoring simulations to the most recent save point if the mutation was lost, or if the allele frequency threshold was not met at the end of the simulation.

To speed up simulations, we applied a scaling factor of $Q = 10$. Scaling divides population sizes ($N$) and event times ($T$) by $Q$, and multiplies the mutation rate µ, recombination rate $r$ and selection coefficient $s$ by $Q$, such that the population genetic parameters $\theta = 4N\mu$, $\rho = 4Nr$, and $Ns$ remain approximately invariant to the applied scaling factor (*Haller and Messer, 2019*). After simulating, we further filtered our AI scenario simulations to exclude those that ended with a minor beneficial allele frequency less than a specific cutoff. We tried two cutoffs—5% and 25%—and present results for both. Rejection sampling within SLiM was not possible at these higher thresholds, as simulations often had low probability of reaching the threshold, particularly for recently introduced mutations. We note that this post-simulation filtering alters the distributions of selection coefficients and times of selection onset used for CNN training.

To investigate Neanderthal gene flow into Europeans, we simulated an out-of-Africa demographic model with a single pulse of Neanderthal gene flow into Europeans but not into African Yoruba (Demographic Model A1, *Figure 1—figure supplement 1*), using a composite of previously published model parameters (*Appendix 3—table 1*). The number of samples to simulate for each population was chosen to match the YRI and CEU panels in the 1000 Genomes dataset (*Auton et al., 2015*), and the two high coverage Neanderthal genomes (*Prüfer et al., 2014*). The two simulated Neanderthals were sampled at times corresponding to the estimated ages of the samples as reported in *Prüfer et al., 2017*. To test model misspecification, we performed an additional 10,000 simulations per simulation scenario on a modified version of this model that also incorporates archaic admixture in Africa (*Ragsdale and Gravel, 2019*) (Demographic Model A2; *Figure 1—figure supplement 1*).

To investigate Denisovan gene flow into Melanesian populations, we simulated an out-of-Africa demographic history incorporating two pulses of Denisovan gene flow (*Malaspinas et al., 2016*; *Jacobs et al., 2019*) implemented as the `PapuansOutOfAfrica_10J19` model in stdpopsim (*Adrion et al., 2020a*). For this demographic model we sampled a single Denisovan and a single Neanderthal (with sampling time of the latter corresponding to the Altai Neanderthal's estimated age). The number of Melanesian samples was chosen to match a subset of the IGDP panel (*Jacobs et al., 2019*). The Baining population of New Britain was excluded at the request of the IGDP data access committee, and we also excluded first-degree relatives, resulting in a total of 139 Melanesian individuals used in the analysis. As this demographic model includes two pulses of Denisovan admixture, we simulated half of our AI simulations to correspond with gene flow from the first pulse, and half from the second pulse.

## Conversion of simulations to genotype matrices

We converted the tree sequence files from the simulations into genotype matrices using the tskit Python API (*Kelleher et al., 2016*). Major alleles (those with sample frequency greater than 0.5 after merging all individuals) were encoded in the matrix as 0, while minor alleles were encoded as 1. In the event of equal counts for both alleles, the major allele was chosen at random. Only sites with a minor allele frequency >5% were retained. For sweep and AI simulations, we excluded the site of the selected mutation.

We note that different simulations result in different numbers of segregating sites, but a constraint for efficient CNN training is that each datum in a batch must have the same dimensions. Existing approaches to solve this problem are to use only a fixed number of segregating sites (*Chan et al., 2018*), to pad the matrix out to the maximum number of observed segregating sites

(*Flagel et al., 2019*), or to use an image-resize function to constrain the size of the input data (*Torada et al., 2019*). Each approach discards spatial information about the local density of segregating sites, although this may be recovered by including an additional vector of inter-site distances as input to the network (*Flagel et al., 2019*).

To obtain the benefits of image resizing (fast training times for reduced sizes and easy application to genomic windows of a fixed size), while avoiding its drawbacks, we chose to resize our input matrices differently, and only along the dimension corresponding to sites. To resize the genomic window to have length $m$, the window was partitioned into $m$ bins, and for each individual haplotype we counted the number of minor alleles observed per bin. Compared with interpolation-based resizing (*Torada et al., 2019*), binning is qualitatively similar, but preserves inter-allele distances and thus the local density of segregating sites. Furthermore, as we do not resize along the dimension corresponding to individuals, this also permits the use of permutation-invariant networks (*Chan et al., 2018*), although we do not pursue that network architecture here.

We report results for $m = 256$, but also tried $m = 32$, 64, and 128 bins. Preliminary results indicated greater training and validation accuracy for CNNs trained with more bins, around 1% difference between both 32 and 64, and 64 and 128, although only marginal improvement for 256 compared with 128 bins. When matching unphased data, we combined genotypes by summing minor allele counts between the chromosomes of each individual. We note that all data were treated as either phased, or unphased, and no mixed phasing was considered.

We then partitioned the resized genotype matrix into submatrices by population. Submatrices were ordered left-to-right according to the donor, recipient, and unadmixed populations respectively. For genotype matrices including both Neanderthals and Denisovans, we placed the non-donor archaic population to the left of the donor. To ensure that a non-permutation-invariant CNN could learn the structure in our data, we sorted the haplotypes (*Flagel et al., 2019*; *Torada et al., 2019*). The resized haplotypes/individuals within each submatrix were ordered left-to-right by decreasing similarity to the donor population, calculated as the Euclidean distance to the average minor-allele density of the donor population (analogous to a vector of the donor allele frequencies). An example (phased) genotype matrix image for an AI simulation is shown in *Figure 1*.

## Conversion of empirical data to genotype matrices

Using bcftools (*Li, 2011*), we performed a locus-wise intersection of the following VCFs: 1000 Genomes (The 1000 *Auton et al., 2015*), IGDP (*Jacobs et al., 2019*), the high coverage Denisovan genome (*Meyer et al., 2012*), and the Altai and Vindija Neanderthal genomes (*Prüfer et al., 2014*). All VCFs corresponded to the GRCh37/hg19 reference sequence. Genotype matrices were constructed by parsing the output of bcftools query over 100 kbp windows, filtering out sites with sample allele frequency <5% or with more than 10% of genotypes missing, then excluding windows with fewer than 20 segregating sites. Each genotype matrix was then resized and sorted as described for simulations. When data were considered to be phased, as for the CEU/YRI populations, we also treated the Neanderthal genotypes as if they were phased according to REF/ALT columns in the VCF. While this is equivalent to random phasing, both high-coverage Neanderthal individuals are highly inbred, so this is unlikely to be problematic in practice.

## CNN model architecture and training

We implemented the CNN model in Keras (*Chollet, 2015*), configured to use the Tensorflow backend (*Abadi et al., 2015*). To save disk space and memory, the input matrices were stored as eight bit integers rather than floating point numbers, and were not mean-centred or otherwise normalised prior to input into the network. We instead made the first layer of our network a batch normalisation layer, which simultaneously converts the input layer to floating point numbers and learns the best normalisation of the data for the network.

The CNN architecture (*Figure 1*) consists of $k$ convolution blocks each comprised of a batch normalisation layer followed by a 2D convolution layer with $2 \times 2$ stride, 16 filters of size $4 \times 4$, and leaky ReLU activation. The $k$ blocks are followed by a single fully-connected output node of size one, with sigmoid activation giving the probability Pr[AI]. We do not include pooling layers, as is common in a CNN architecture (e.g. *Torada et al., 2019*), and instead use a $2 \times 2$ stride size to reduce the output size of successive blocks (*Springenberg et al., 2015*). This is computationally cheaper and

had no observable difference in network performance. We sought to maximise the depth of the network, but the size of the input matrix constrains the maximum number of blocks in the network due to successive halving of the dimensionality in each block. For $m = 256$ resizing bins, we used $k = 7$ blocks.

We partitioned 100,000 independent simulations for each of the three selection scenarios into training and validation sets (approximate 90%/10% split). The hyperparameters and network architecture were tuned on a smaller preliminary set of simulations that did not vary the selection coefficient or time of onset of selection, so we chose not to split the simulations into a third 'test' set when evaluating the models trained on our final simulations. The model was trained for three epochs, with model weights updated after batches of 64, using the Adam optimiser and cross-entropy for the loss function. We evaluated model fit by inspecting loss and accuracy terms at the end of training (*Appendix 2—table 1*). Preliminary analyses indicated three epochs were sufficient for approximate convergence between training and validation metrics, but we did not observe divergence (likely indicating overfitting) even when training for additional epochs.

## Comparison to other methods

We converted our simulated tree sequences to VolcanoFinder (*Setter et al., 2020*) input (a per-locus allele counts file, and a frequency-spectrum input file). One frequency-spectrum input file was created for each distinct demographic model, obtained by averaging over all neutral-scenario simulations. VolcanoFinder was run following examples in the manual, using 800 evenly-spaced genomic bins (-G 800), and taking the maximum value for the likelihood-ratio test statistic (LRT) as the summary statistic for that simulation. VolcanoFinder further requires a value for the divergence between the donor and recipient populations, which it can estimate by doing a grid search for the value which maximises the LRT. However, this is more computationally intensive than providing a value, so we obtained a value by grid search for a small sample of our simulations for each of Demographic Models A1 and B, and used the most frequently observed value (-D 0.001075 for Demographic Model A1 and A2, and -D 0.001465 for Demographic Model B).

Additional AI-related summary statistics were chosen based on *Racimo et al., 2017b* and calculated on the simulated tree sequences. The $f_d$ statistic was implemented from the description in *Martin et al., 2015* and the remaining statistics were implemented from their description in *Racimo et al., 2017b*. Summary statistics (including the VolcanoFinder LRT) were obtained for 600,000 simulations (100,000 for each of three simulation scenarios, for each of Demographic Models A1 and B). We calculated p-values by comparing each statistic to the null distribution that was obtained from the neutral simulation scenarios.

## Calibration

For a well calibrated output, we expect proportion $x$ of the output probabilities with Pr[AI] $\sim x$ to be true positives. It has been noted elsewhere (*Guo et al., 2017*) that CNNs may produce improperly calibrated probabilities. However, even if the probabilities are calibrated with respect to the validation dataset (which has even class ratios), this is unlikely to hold for empirical data, as the relative ratios of AI versus not-AI windows in the genome are very skewed.

We tested three calibration methods: beta regression (*Kull et al., 2017*), isotonic regression (*Chakravarti, 1989*), and *Platt, 1999* scaling. To calibrate our CNN output, we first resampled our training dataset to the desired class ratios. We then fit each calibrator to predict the true class in the resampled training dataset from the CNN prediction for the resampled training dataset. To assess the calibration procedure, we inspected reliability plots for our calibrated and uncalibrated predictions, as evaluated with a resampled validation dataset (*Figure 4—figure supplement 1*, *Figure 4—figure supplement 2*, *Figure 4—figure supplement 3*, *Figure 4—figure supplement 4*). We also checked if the sum of the residuals was normally distributed, following the approach of *Turner et al., 2019*. Both beta calibration and isotonic regression gave well-calibrated probabilities compared with uncalibrated model outputs, and for our predictions on empirical data we chose to apply beta calibration due to its relative simplicity.

## Saliency maps

We computed average saliency maps, by averaging over a set of input-specific saliency maps that were calculated for a set of 300 simulated genotype matrices for each simulated scenario. The input-specific saliency maps were calculated using `tf-keras-vis` v0.5.5 (*Kubota, 2020*) configured to use 'vanilla' saliency maps. A sharper image was obtained by exchanging the CNN output layer's sigmoid activation with linear activation, as recommended in the `tf-keras-vis` documentation. For the 'vanilla' saliency option, the image-specific class saliency is calculated by computing the gradient of a network's output with respect to a single input. The exact details of how the saliency is calculated via propagation through a neural network can be found in *Simonyan et al., 2014*, who offer this interpretation: '[T]he magnitude of the derivative indicates which pixels need to be changed the least to affect the class score the most'.

## Application of trained CNN to empirical datasets

We show Manhattan plots where each data point is a 100 kbp window that moves along the genome in steps of size 20 kbp. Gene annotations were extracted from the Ensembl release 87 GFF3 file (with GRCh37/hg19 coordinates), obtained via ensembl's ftp server. We extracted the columns with source='ensembl_havana' and type='gene', and report the genes which intersected with the 30 top ranking CNN predictions or a 100 kbp flanking region. Adjacent regions were merged together prior to intersection, so that genes were reported only once.

## Compute resources

All simulations and results reported here were obtained on an compute server with two Intel Xeon 6248 CPUs (80 cores total), 768 GB RAM, and five NVIDIA Tesla T4 GPUs. 300,000 SLiM simulations took approximately 80 weeks of CPU time for each of Demographic Model A1 and B. Each simulation executes independently, and is readily distributed across cores or compute nodes. This produced 450 GB of tree sequence files. The resized genotype matrices were compressed into a Zarr cache (*Zarr Development Team, 2020*) with size 2.8 GB, for faster loading. Training a single CNN on one GPU took approximately 15 min, or 60 CPU hours for an equivalent CPU-only training procedure. We did not attempt to optimise memory usage, and thus all data were loaded into memory, requiring approximately 120 GB RAM during training. Predicting AI for all genomic windows on an empirical dataset (22 single-chromosome BCF files) took 1 CPU hour. However, our prediction pipeline uses multiprocessing and efficiently scales to 80 cores.

## Code availability

The source code for performing simulations, training and evaluating a CNN, and applying a CNN to empirical VCF data, were developed in a new Python application called genomatnn, available at *Gower, 2021*.

## Acknowledgements

We thank Andrew Kern, Martin Sikora, Flora Jay and Anders Albrechtsen, as well as three anonymous reviewers, and the members of the Racimo group and the PopSim consortium, for helpful advice and discussions. We also thank Murray Cox and Georgi Hudjashov for facilitating access to the IGDP data. Funding FR and GG were supported by a Villum Fonden Young Investigator award to FR (project no. 00025300). MF was supported by a Leverhulme Research Project grant (RPG-2018–208) and the Imperial College European Partners Fund. FR was also supported by a Lundbeck-fonden grant (R302-2018-2155) and a Novo Nordisk Fonden grant (NNF18SA0035006) to the Geo-Genetics Centre. The funding sources were not involved in study design, data collection and interpretation, or the decision to submit the work for publication.

## Additional information

### Funding

| Funder | Grant reference number | Author |
|---|---|---|
| Villum Fonden | 00025300 | Fernando Racimo |
| Leverhulme Trust | RPG-2018-208 | Matteo Fumagalli |
| Lundbeckfonden | R302-2018-21555 | Fernando Racimo |
| Novo Nordisk Fonden | NNF18SA0035006 | Fernando Racimo |

The funders had no role in study design, data collection and interpretation, or the decision to submit the work for publication.

### Author contributions
Graham Gower, Software, Formal analysis, Investigation, Visualization, Methodology, Writing - original draft, Writing - review and editing; Pablo Iáñez Picazo, Software, Formal analysis, Visualization, Methodology, Writing - review and editing; Matteo Fumagalli, Conceptualization, Writing - review and editing; Fernando Racimo, Conceptualization, Supervision, Funding acquisition, Methodology, Writing - original draft, Writing - review and editing

### Author ORCIDs
Graham Gower (ID) https://orcid.org/0000-0002-6197-3872
Pablo Iáñez Picazo (ID) https://orcid.org/0000-0001-7174-3264
Matteo Fumagalli (ID) http://orcid.org/0000-0002-4084-2953
Fernando Racimo (ID) https://orcid.org/0000-0002-5025-2607

### Decision letter and Author response
Decision letter https://doi.org/10.7554/eLife.64669.sa1
Author response https://doi.org/10.7554/eLife.64669.sa2

## Additional files

### Supplementary files
• Transparent reporting form

### Data availability
Source code is available from https://github.com/grahamgower/genomatnn/.

The following datasets were generated:

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

# Appendix 1

**Appendix 1—table 1.** Top ranking gene candidates corresponding to Neanderthal AI in Europeans. We show genes which overlap, or are within 100 kbp of, the 30 highest ranked 100 kbp intervals. Adjacent intervals have been merged. The CNN was trained using only AI simulations with selected mutation having allele frequency >5%, and subsequently calibrated with resampled neutral:sweep:AI ratios of 1:0.1:0.02 .c

| Chrom | Start | End | Genes |
|---|---|---|---|
| 1 | 39420001 | 39520000 | RRAGC; MYCBP; GJA9; RHBDL2; AKIRIN1; NDUFS5; MACF1 |
| 2 | 159880001 | 160280000 | TANC1; WDSUB1; BAZ2B |
| 2 | 180060001 | 180160000 | SESTD1 |
| 2 | 227800001 | 227900000 | RHBDD1; COL4A4 |
| 2 | 238820001 | 238960000 | LRRFIP1; RBM44; RAMP1; UBE2F; SCLY; ESPNL; KLHL30 |
| 3 | 114500001 | 114600000 | ZBTB20 |
| 5 | 57960001 | 58060000 | RAB3C |
| 6 | 28160001 | 28380000 | ZSCAN16-AS1; ZSCAN16; ZKSCAN8; ZSCAN9; ZKSCAN4; NKAPL; PGBD1; ZSCAN31; ZKSCAN3; ZSCAN12; ZSCAN23; GPX6 |
| 8 | 17060001 | 17160000 | MICU3; ZDHHC2; CNOT7; VPS37A; MTMR7 |
| 8 | 91840001 | 91940000 | TMEM64; NECAB1; TMEM55A |
| 9 | 16700001 | 16860000 | BNC2 |
| 10 | 11800001 | 11900000 | ECHDC3; PROSER2; UPF2 |
| 11 | 37740001 | 37840000 | |
| 19 | 20260001 | 20360000 | ZNF90; ZNF486 |
| 19 | 33580001 | 33700000 | RHPN2; GPATCH1; WDR88; LRP3; SLC7A10 |
| 20 | 14340001 | 14440000 | MACROD2; FLRT3 |

**Appendix 1—table 2.** Top ranking gene candidates corresponding to Denisovan AI in Melanesians. We show genes which overlap, or are within 100 kbp of, the 30 highest ranked 100 kbp intervals. Adjacent intervals have been merged. The CNN was trained using only AI simulations with selected mutation having allele frequency >5%, and subsequently calibrated with resampled neutral:sweep:AI ratios of 1:0.1:0.02.

| Chrom | Start | End | Genes |
|---|---|---|---|
| 1 | 2880001 | 2980000 | ACTRT2; LINC00982; PRDM16 |
| 1 | 220080001 | 220180000 | SLC30A10; EPRS; BPNT1; IARS2 |
| 2 | 221040001 | 221140000 | |
| 3 | 15400001 | 15500000 | SH3BP5; METTL6; EAF1; COLQ |
| 4 | 41960001 | 42100000 | TMEM33; DCAF4L1; SLC30A9; BEND4 |
| 5 | 135440001 | 135540000 | TGFBI; SMAD5-AS1; SMAD5; TRPC7 |
| 6 | 81980001 | 82120000 | FAM46A |
| 7 | 121160001 | 121260000 | |
| 9 | 95500001 | 95600000 | IPPK; BICD2; ZNF484 |
| 10 | 59660001 | 59760000 | |
| 12 | 80780001 | 80880000 | OTOGL; PTPRQ |

*Continued on next page*

Appendix 1—table 2 continued

| Chrom | Start | End | Genes |
|---|---|---|---|
| 12 | 84620001 | 84740000 | |
| 14 | 57620001 | 57760000 | EXOC5; AP5M1; NAA30 |
| 17 | 29480001 | 29720000 | NF1; OMG; EVI2B; EVI2A; RAB11FIP4 |
| 18 | 38180001 | 38320000 | |
| 20 | 54340001 | 54440000 | |

# Appendix 2

**Appendix 2—table 1.** Loss and accuracy for CNNs after training for three epochs, as reported by Keras/Tensorflow, for the training and validation datasets.
Binary cross-entropy was used for the loss function.

| Demographic model | Hyperparameters | Training | | Validation | |
|---|---|---|---|---|---|
| | | **Loss** | **Accuracy** | **Loss** | **Accuracy** |
| A1 | AF>0.05 | 0.1592 | 0.9458 | 0.1618 | 0.9468 |
| A1 | AF>0.25 | 0.1224 | 0.9585 | 0.1265 | 0.9578 |
| A1 | AF>0.25; unphased | 0.1347 | 0.9537 | 0.1368 | 0.9530 |
| B | AF>0.05; unphased | 0.3415 | 0.8439 | 0.3441 | 0.8439 |
| B | AF>0.25; unphased | 0.3546 | 0.8372 | 0.3583 | 0.8376 |

# Appendix 3

**Appendix 3—table 1.** Parameter values used for simulating Demographic Model A1.
A Demes-format YAML file for each demographic model is available from the genomatnn git repository.

| Parameter | Description | Value | Units | Source |
|---|---|---|---|---|
| $N_{Anc}$ | ancestral pop. size | 18500 | | *Kuhlwilm et al., 2016* |
| $N_{Nea}$ | Neanderthal pop. size | 3400 | | *Kuhlwilm et al., 2016* |
| $N_{YRI}$ | YRI pop. size | 27600 | | *Kuhlwilm et al., 2016* |
| $N_{CEU0}$ | CEU bottleneck pop. size | 1080 | | *Ragsdale and Gravel, 2019* |
| $N_{CEU1}$ | CEU growth-start pop. size | 1450 | | *Ragsdale and Gravel, 2019* |
| $N_{CEU2}$ | CEU current pop. size | 13377 | | |
| $r_{CEU}$ | CEU growth rate | 0.00202 | | *Ragsdale and Gravel, 2019* |
| $T_{CEU2}$ | CEU time at growth start | 31.9 | kya | *Ragsdale and Gravel, 2019* |
| $T_0$ | Nea/other split time | 550 | kya | *Prüfer et al., 2017* |
| $T_1$ | CEU/YRI split time | 65.7 | kya | *Ragsdale and Gravel, 2019* |
| $T_2$ | time of Nea → CEU gene flow | 55 | kya | *Prüfer et al., 2017* |
| $g$ | generation time | 29 | years | *Prüfer et al., 2017* |
| $\alpha$ | Nea → CEU admixture proportion | 2.25 | | *Prüfer et al., 2017* |
| $T_{Altai}$ | sampling time | 115 | kya | *Prüfer et al., 2017* |
| $T_{Vindija}$ | sampling time | 55 | kya | *Prüfer et al., 2017* |
| $n_{Nean}$ | sample size | 2 | diploid individuals | |
| $n_{Afr}$ | sample size | 108 | diploid individuals | |
| $n_{Eur}$ | sample size | 99 | diploid individuals | |
| $s$ | selection coefficient | $10^{\text{Unif}(-4,-1)}$ | | |
| $T_{sel1}$ | selection onset (sweep) | $\text{Unif}(1, T_1)$ | kya | |
| $T_{mut1}$ | mutation (sweep) | $\text{Unif}(T_{sel1}, T_1)$ | kya | |
| $T_{sel2}$ | selection onset (AI) | $\text{Unif}(1, T_2)$ | kya | |
| $T_{mut2}$ | mutation (AI) | $\text{Unif}(T_2, T_0)$ | kya | |

## Appendix 4

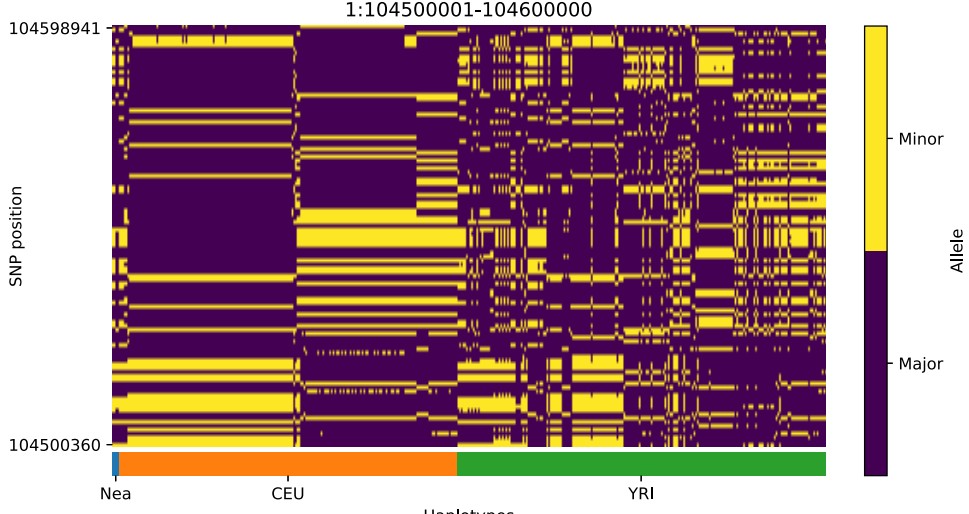

**Appendix 4—figure 1.** Haplotype plot for the candidate region chr1:104500001–104600000 in the Neanderthal-into-European AI scan. Bright yellow indicates minor allele, dark blue indicates major allele. Haplotypes within populations are sorted left-to-right by similarity to Neanderthals.

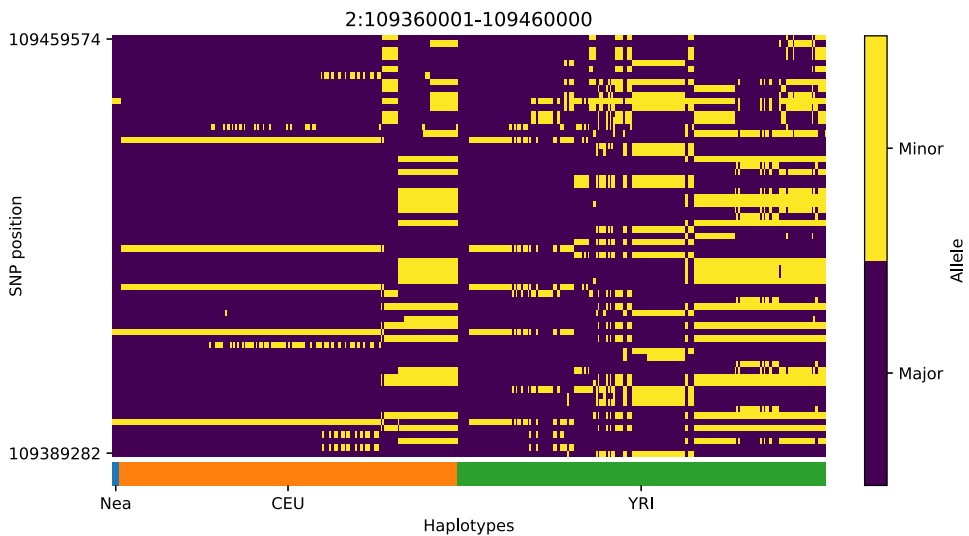

**Appendix 4—figure 2.** Haplotype plot for the candidate region chr2:109360001–109460000 in the Neanderthal-into-European AI scan. Bright yellow indicates minor allele, dark blue indicates major allele. Haplotypes within populations are sorted left-to-right by similarity to Neanderthals.

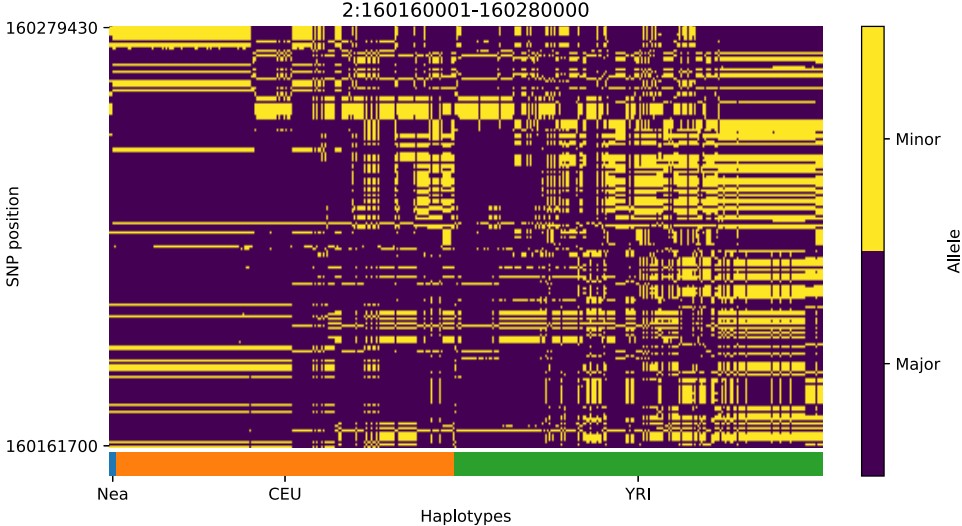

**Appendix 4—figure 3.** Haplotype plot for the candidate region chr2:160160001–160280000 in the Neanderthal-into-European AI scan. Bright yellow indicates minor allele, dark blue indicates major allele. Haplotypes within populations are sorted left-to-right by similarity to Neanderthals.

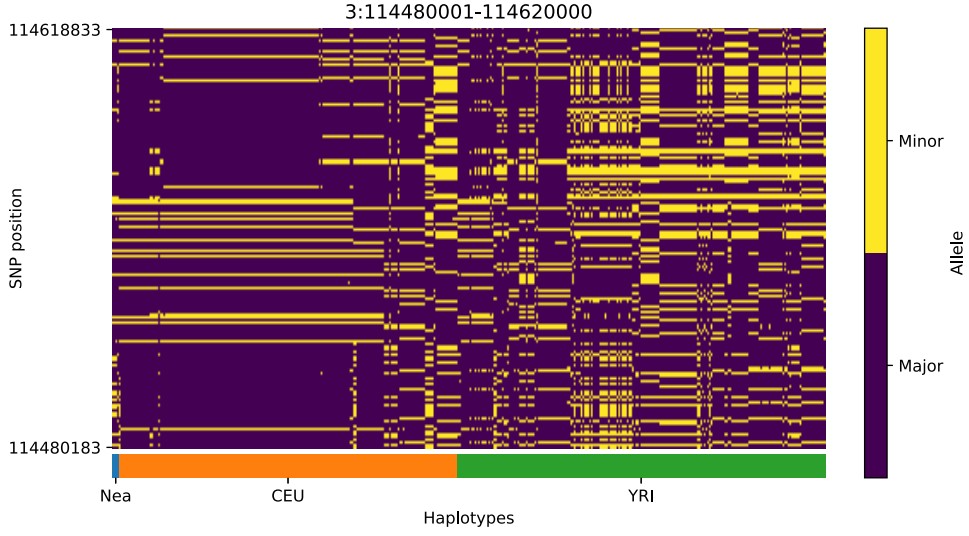

**Appendix 4—figure 4.** Haplotype plot for the candidate region chr3:114480001–114620000 in the Neanderthal-into-European AI scan. Bright yellow indicates minor allele, dark blue indicates major allele. Haplotypes within populations are sorted left-to-right by similarity to Neanderthals.

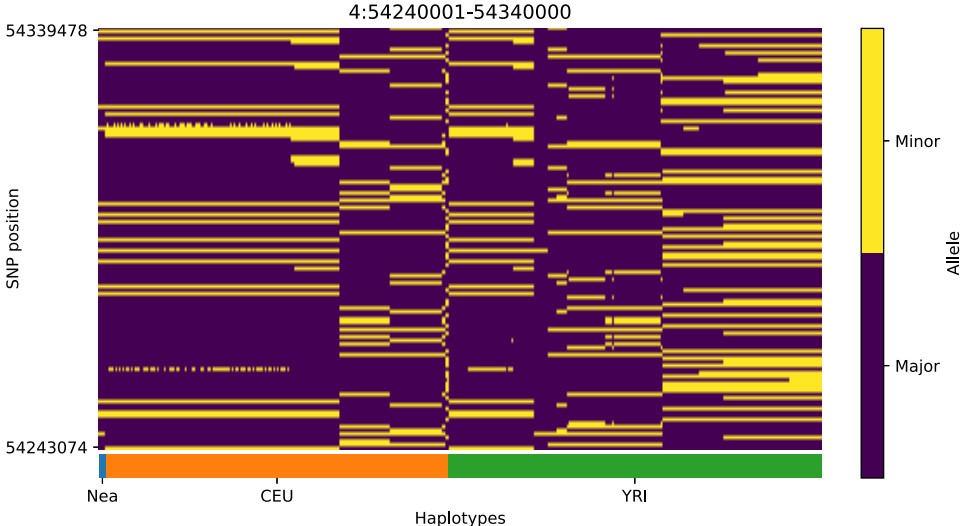

**Appendix 4—figure 5.** Haplotype plot for the candidate region chr4:54240001–54340000 in the Neanderthal-into-European AI scan. Bright yellow indicates minor allele, dark blue indicates major allele. Haplotypes within populations are sorted left-to-right by similarity to Neanderthals.

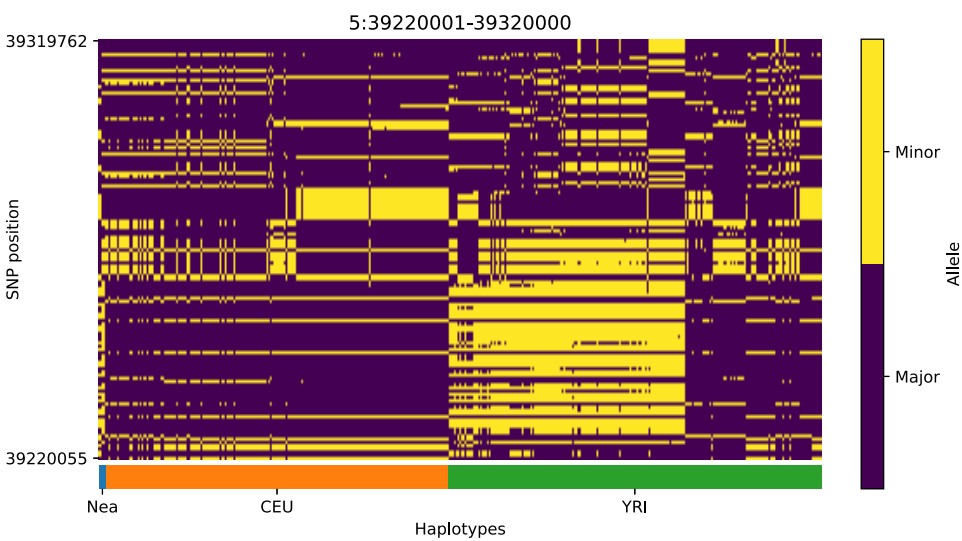

**Appendix 4—figure 6.** Haplotype plot for the candidate region chr5:39220001–39320000 in the Neanderthal-into-European AI scan. Bright yellow indicates minor allele, dark blue indicates major allele. Haplotypes within populations are sorted left-to-right by similarity to Neanderthals.

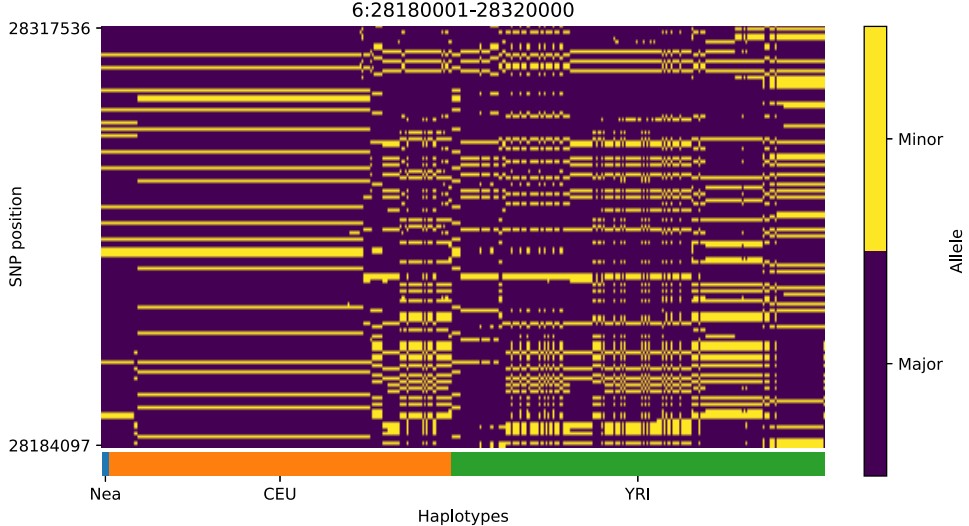

**Appendix 4—figure 7.** Haplotype plot for the candidate region chr6:28180001–28320000 in the Neanderthal-into-European AI scan. Bright yellow indicates minor allele, dark blue indicates major allele. Haplotypes within populations are sorted left-to-right by similarity to Neanderthals.

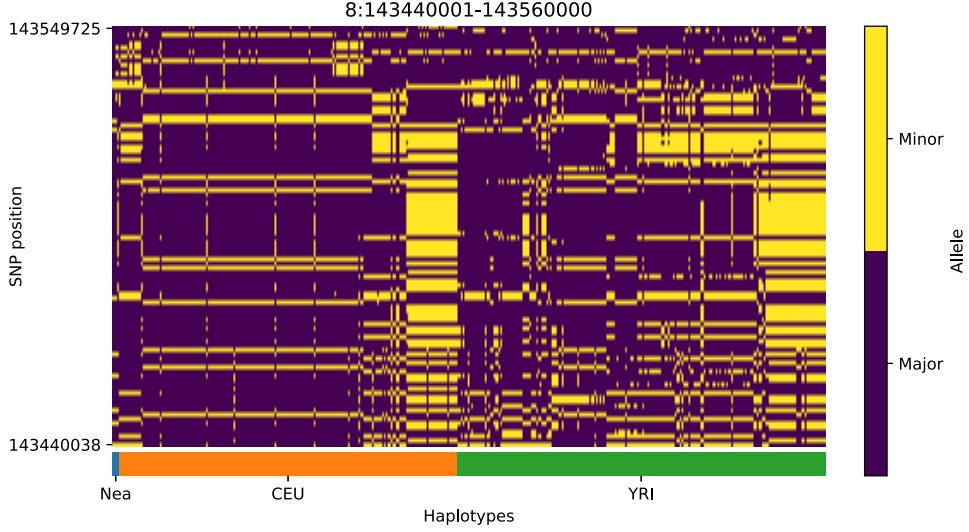

**Appendix 4—figure 8.** Haplotype plot for the candidate region chr8:143440001–143560000 in the Neanderthal-into-European AI scan. Bright yellow indicates minor allele, dark blue indicates major allele. Haplotypes within populations are sorted left-to-right by similarity to Neanderthals.

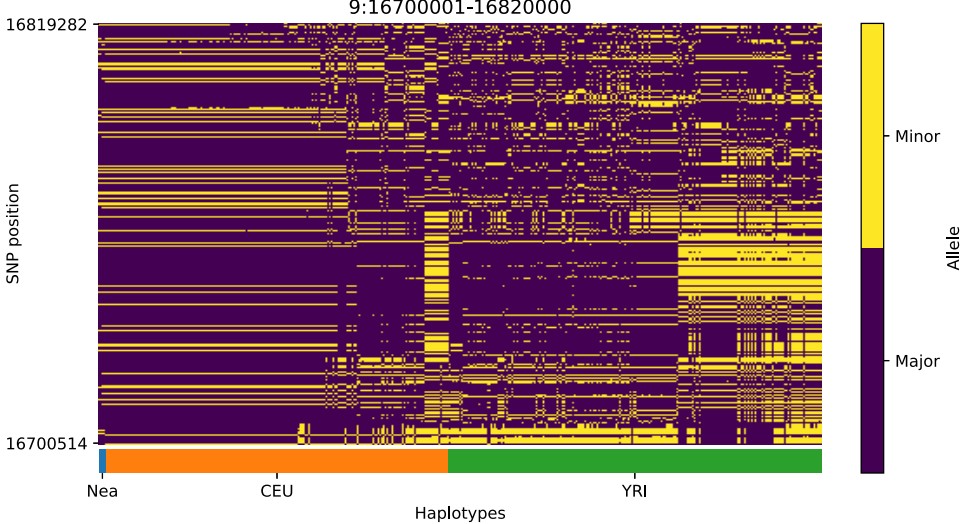

**Appendix 4—figure 9.** Haplotype plot for the candidate region chr9:16700001–16820000 in the Neanderthal-into-European AI scan. Bright yellow indicates minor allele, dark blue indicates major allele. Haplotypes within populations are sorted left-to-right by similarity to Neanderthals.

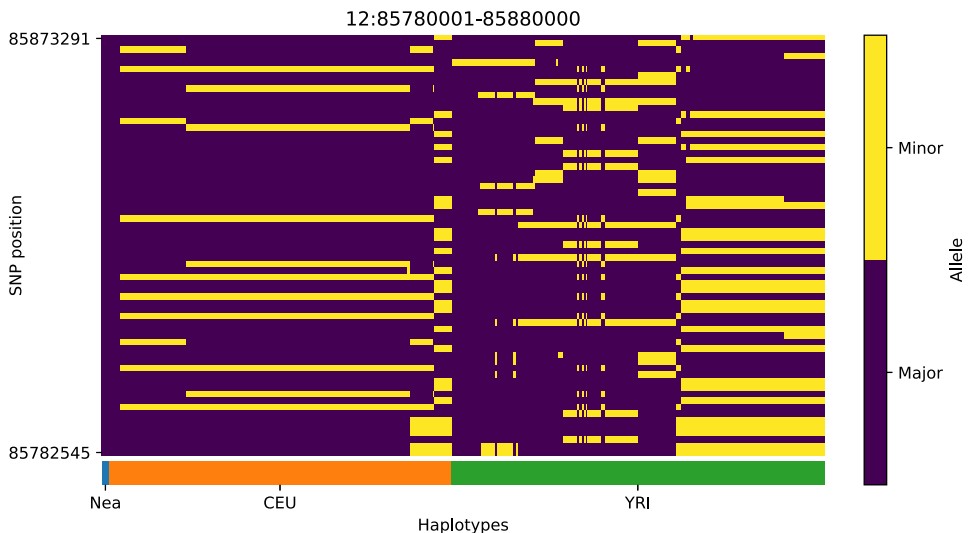

**Appendix 4—figure 10.** Haplotype plot for the candidate region chr12:85780001–85880000 in the Neanderthal-into-European AI scan. Bright yellow indicates minor allele, dark blue indicates major allele. Haplotypes within populations are sorted left-to-right by similarity to Neanderthals.

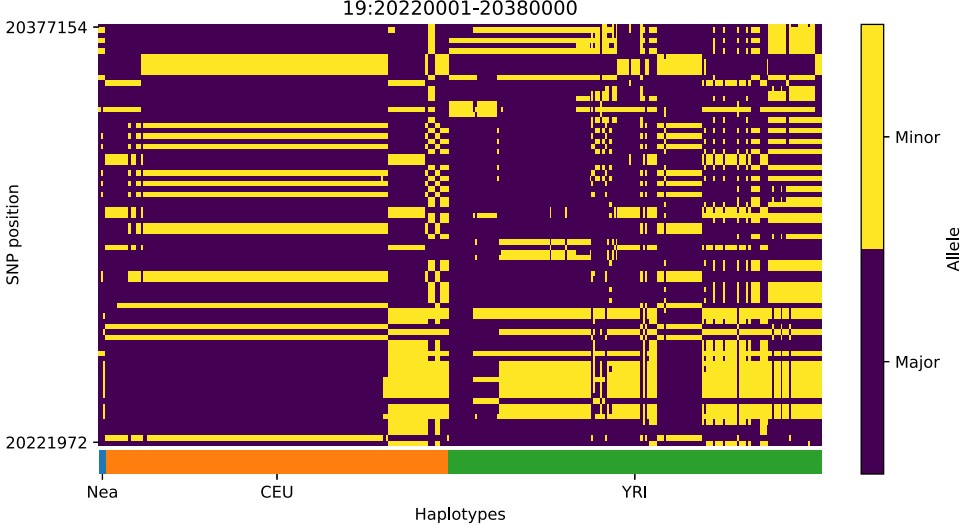

**Appendix 4—figure 11.** Haplotype plot for the candidate region chr19:20220001–20380000 in the Neanderthal-into-European AI scan. Bright yellow indicates minor allele, dark blue indicates major allele. Haplotypes within populations are sorted left-to-right by similarity to Neanderthals.

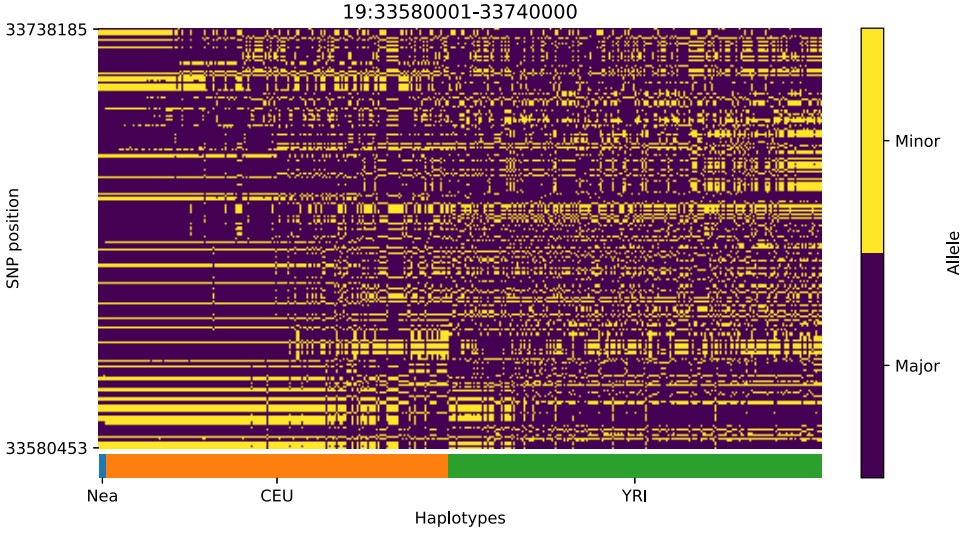

**Appendix 4—figure 12.** Haplotype plot for the candidate region chr19:33580001–33740000 in the Neanderthal-into-European AI scan. Bright yellow indicates minor allele, dark blue indicates major allele. Haplotypes within populations are sorted left-to-right by similarity to Neanderthals.

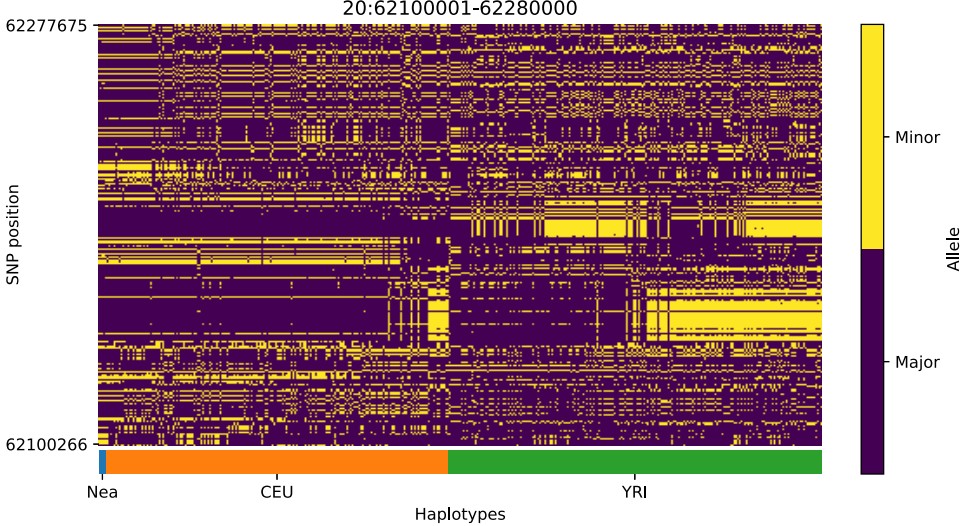

**Appendix 4—figure 13.** Haplotype plot for the candidate region chr20:62100001–62280000 in the Neanderthal-into-European AI scan. Bright yellow indicates minor allele, dark blue indicates major allele. Haplotypes within populations are sorted left-to-right by similarity to Neanderthals.

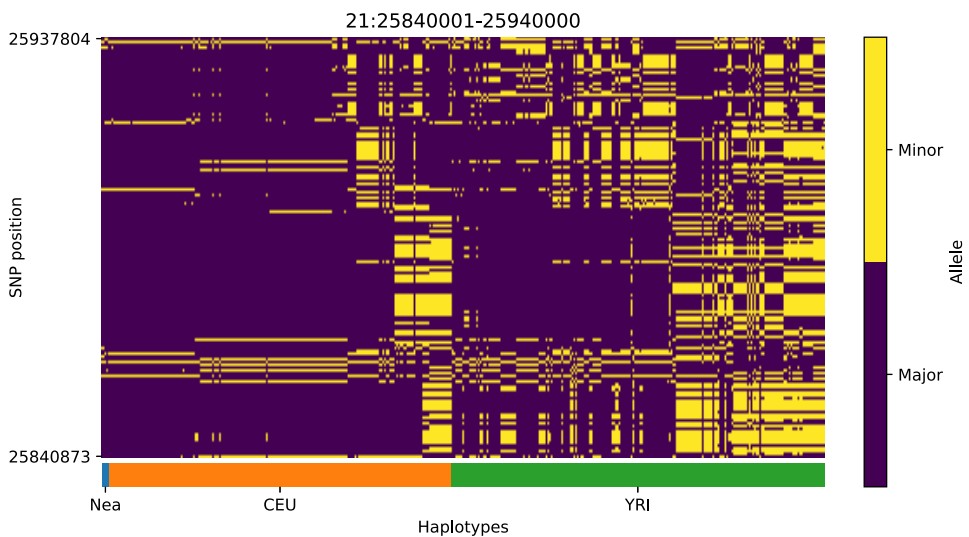

**Appendix 4—figure 14.** Haplotype plot for the candidate region chr21:25840001–25940000 in the Neanderthal-into-European AI scan. Bright yellow indicates minor allele, dark blue indicates major allele. Haplotypes within populations are sorted left-to-right by similarity to Neanderthals.

## Appendix 5

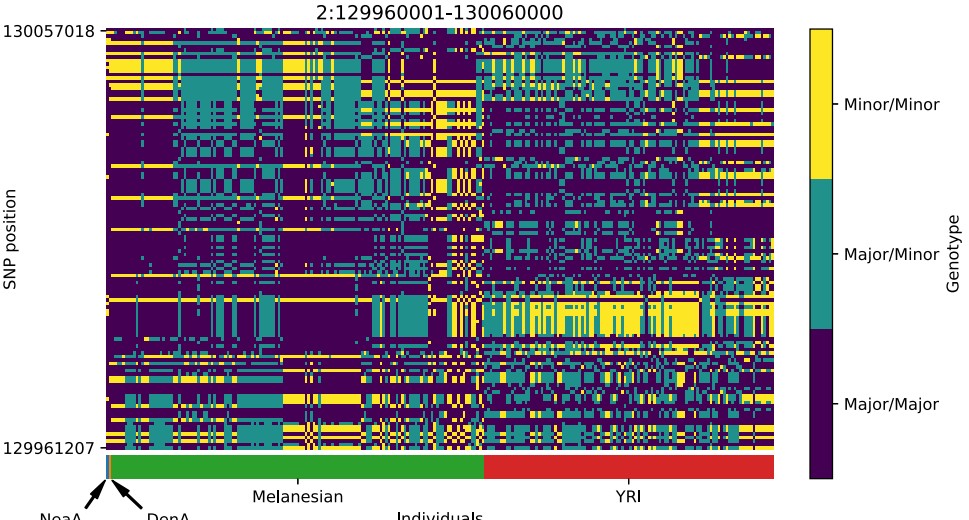

**Appendix 5—figure 1.** Genotype plot for the candidate region chr2:129960001–130060000 in the Denisovan-into-Melanesian AI scan. Dark blue = homozygote major allele, light blue = heterozygote, yellow = homozygote minor allele. Genotypes within populations are sorted left-to-right by similarity to the Denisovan.

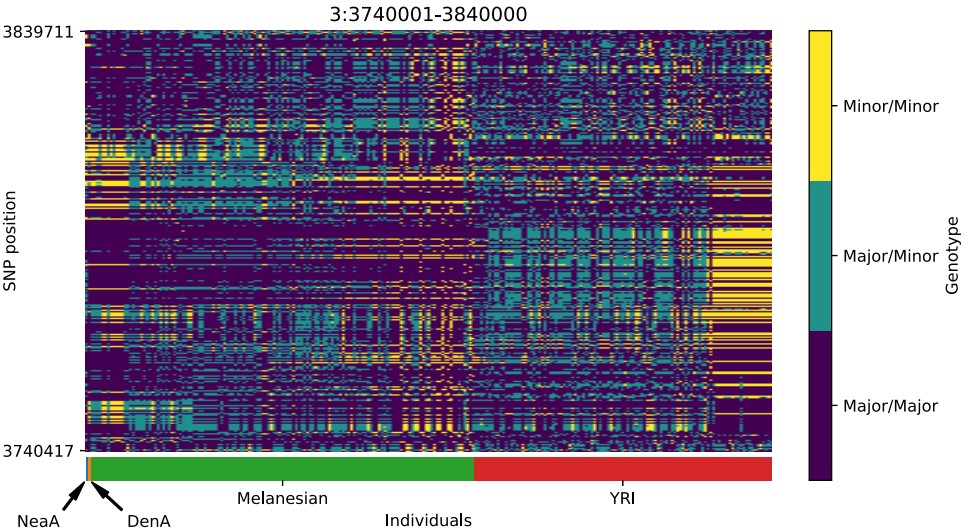

**Appendix 5—figure 2.** Genotype plot for the candidate region chr3:3740001–3840000 in the Denisovan-into-Melanesian AI scan. Dark blue = homozygote major allele, light blue = heterozygote, yellow = homozygote minor allele. Genotypes within populations are sorted left-to-right by similarity to the Denisovan.

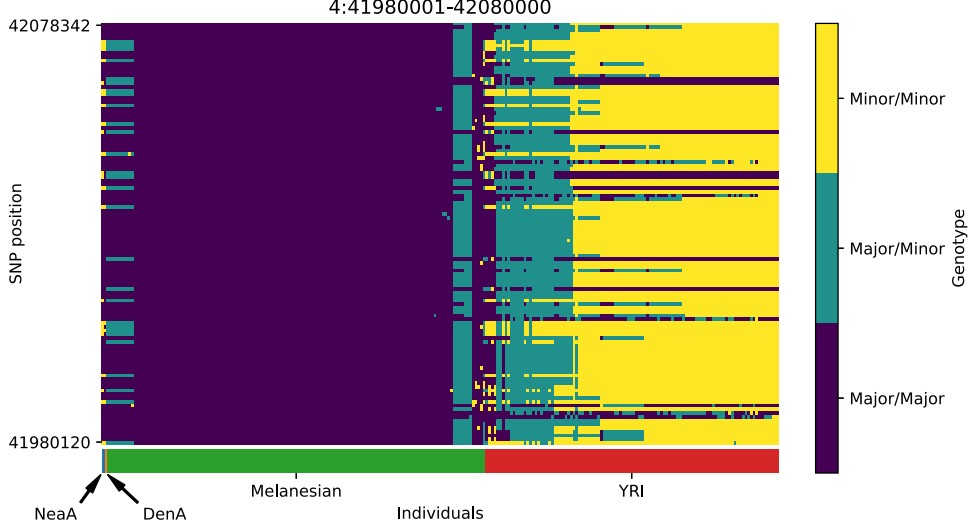

**Appendix 5—figure 3.** Genotype plot for the candidate region chr4:41980001–42080000 in the Denisovan-into-Melanesian AI scan. Dark blue = homozygote major allele, light blue = heterozygote, yellow = homozygote minor allele. Genotypes within populations are sorted left-to-right by similarity to the Denisovan.

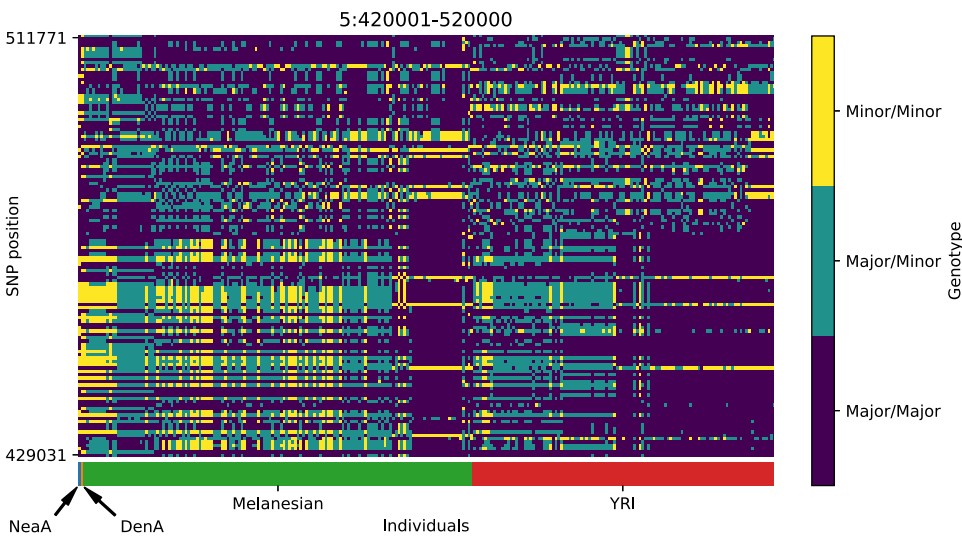

**Appendix 5—figure 4.** Genotype plot for the candidate region chr5:420001–520000 in the Denisovan-into-Melanesian AI scan. Dark blue = homozygote major allele, light blue = heterozygote, yellow = homozygote minor allele. Genotypes within populations are sorted left-to-right by similarity to the Denisovan.

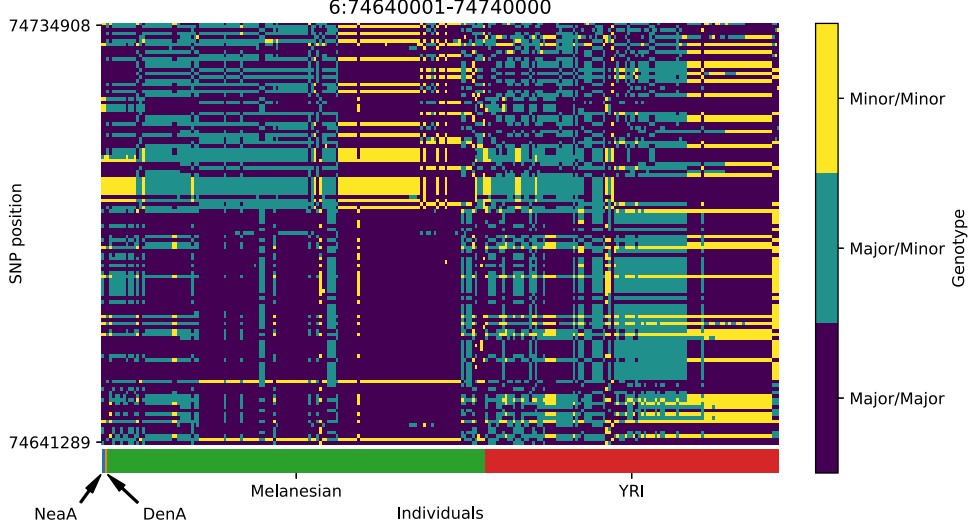

**Appendix 5—figure 5.** Genotype plot for the candidate region chr6:74640001–74740000 in the Denisovan-into-Melanesian AI scan. Dark blue = homozygote major allele, light blue = heterozygote, yellow = homozygote minor allele. Genotypes within populations are sorted left-to-right by similarity to the Denisovan.

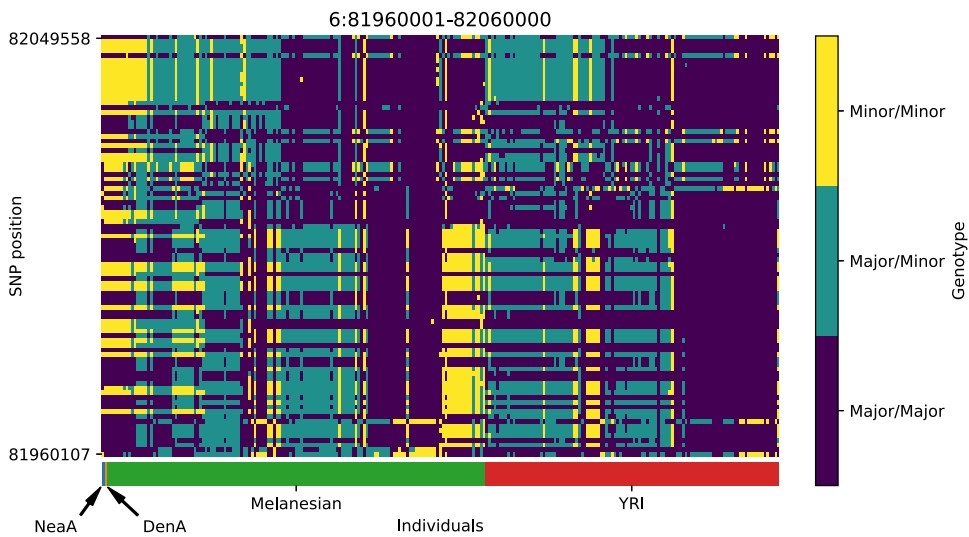

**Appendix 5—figure 6.** Genotype plot for the candidate region chr6:81960001–82060000 in the Denisovan-into-Melanesian AI scan. Dark blue = homozygote major allele, light blue = heterozygote, yellow = homozygote minor allele. Genotypes within populations are sorted left-to-right by similarity to the Denisovan.

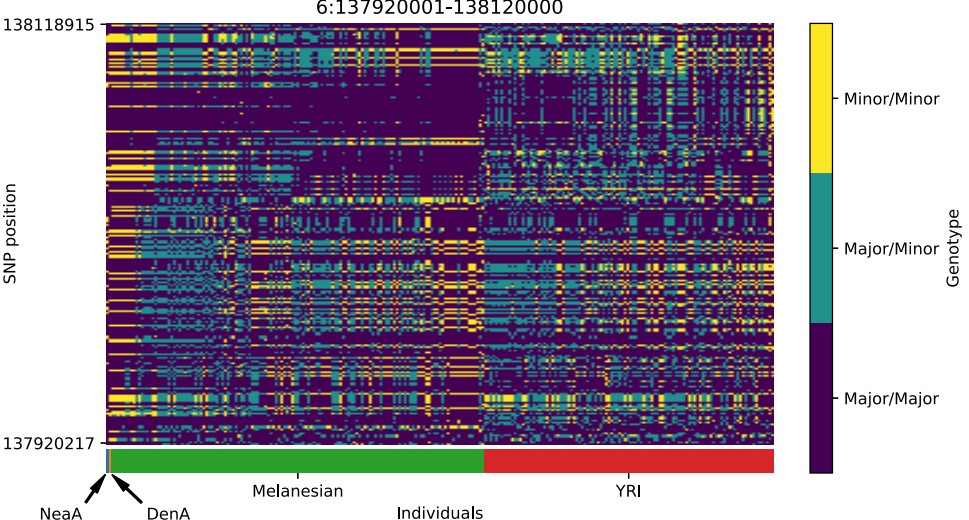

**Appendix 5—figure 7.** Genotype plot for the candidate region chr6:137920001–138120000 in the Denisovan-into-Melanesian AI scan. Dark blue = homozygote major allele, light blue = heterozygote, yellow = homozygote minor allele. Genotypes within populations are sorted left-to-right by similarity to the Denisovan.

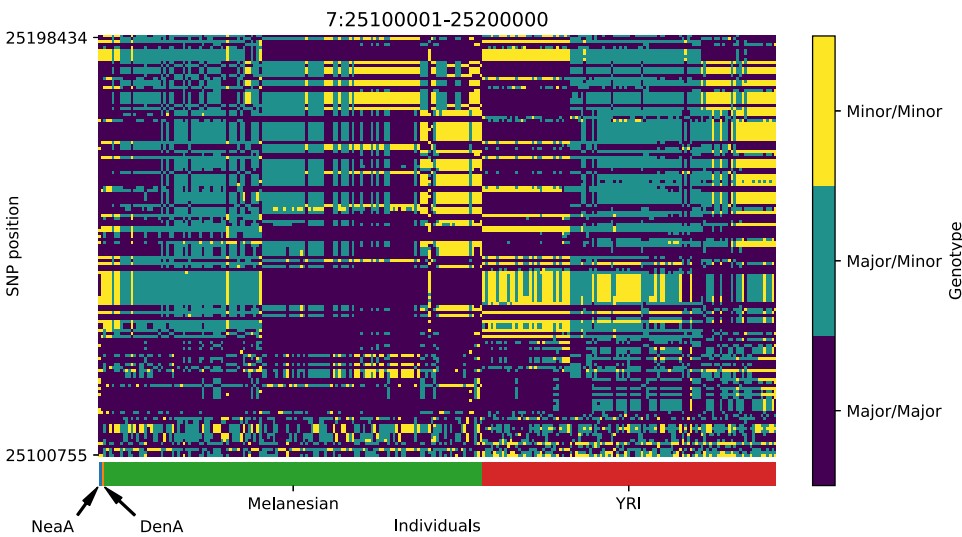

**Appendix 5—figure 8.** Genotype plot for the candidate region chr7:25100001–25200000 in the Denisovan-into-Melanesian AI scan. Dark blue = homozygote major allele, light blue = heterozygote, yellow = homozygote minor allele. Genotypes within populations are sorted left-to-right by similarity to the Denisovan.

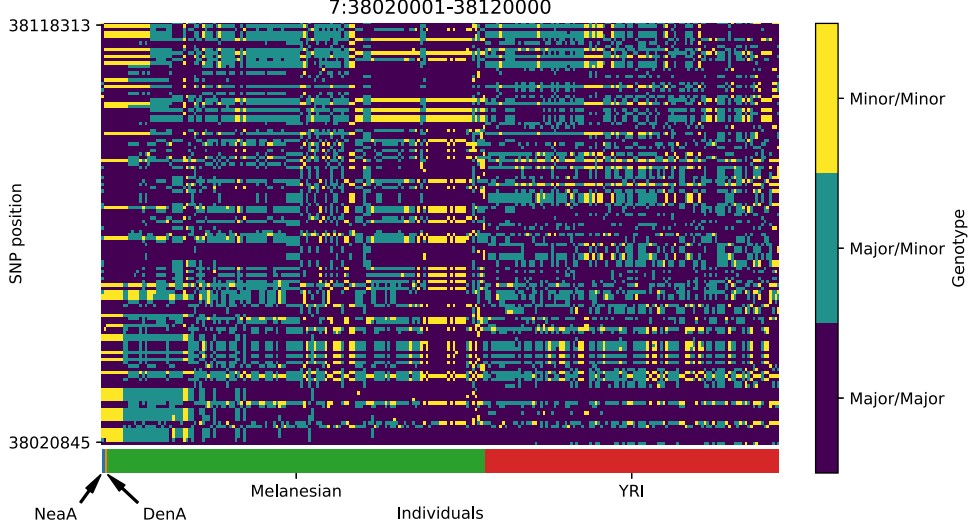

**Appendix 5—figure 9.** Genotype plot for the candidate region chr7:38020001–38120000 in the Denisovan-into-Melanesian AI scan. Dark blue = homozygote major allele, light blue = heterozygote, yellow = homozygote minor allele. Genotypes within populations are sorted left-to-right by similarity to the Denisovan.

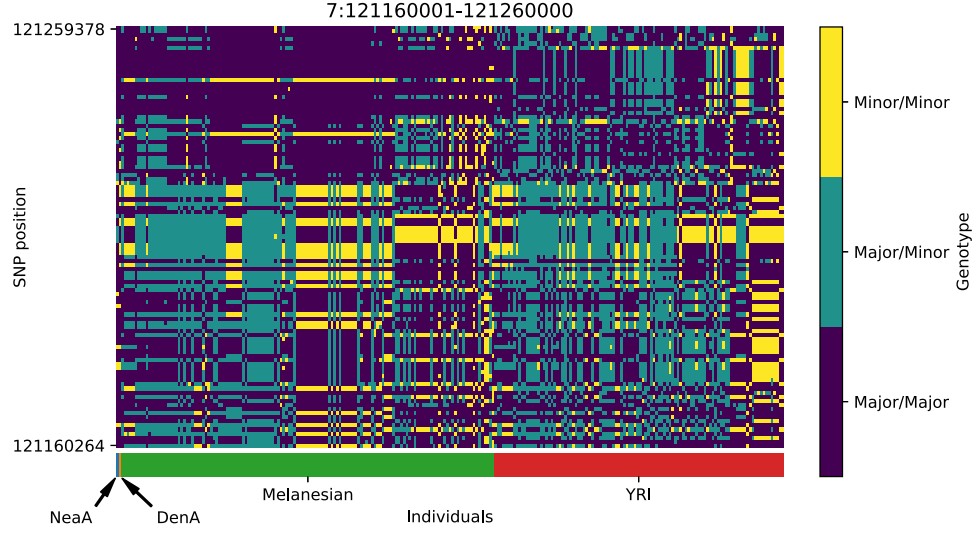

**Appendix 5—figure 10.** Genotype plot for the candidate region chr7:121160001–121260000 in the Denisovan-into-Melanesian AI scan. Dark blue = homozygote major allele, light blue = heterozygote, yellow = homozygote minor allele. Genotypes within populations are sorted left-to-right by similarity to the Denisovan.

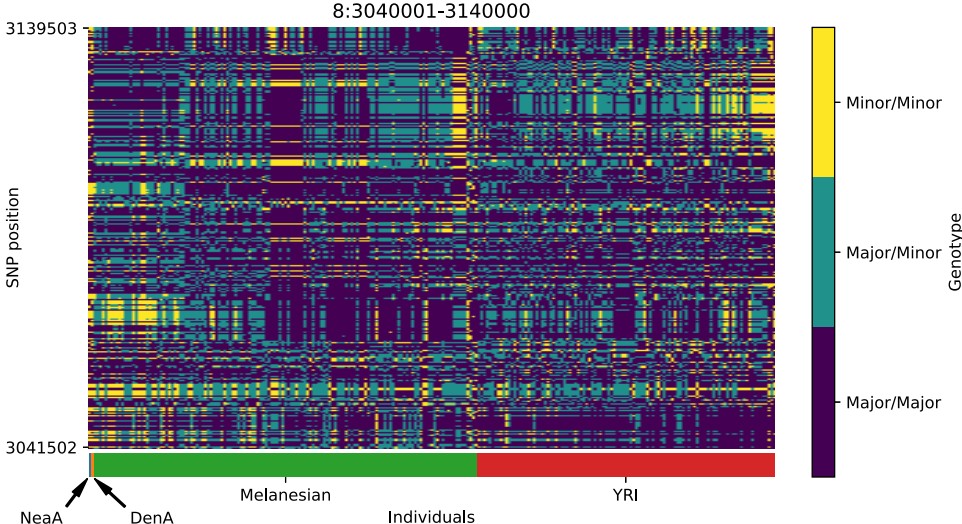

**Appendix 5—figure 11.** Genotype plot for the candidate region chr8:3040001–3140000 in the Denisovan-into-Melanesian AI scan. Dark blue = homozygote major allele, light blue = heterozygote, yellow = homozygote minor allele. Genotypes within populations are sorted left-to-right by similarity to the Denisovan.

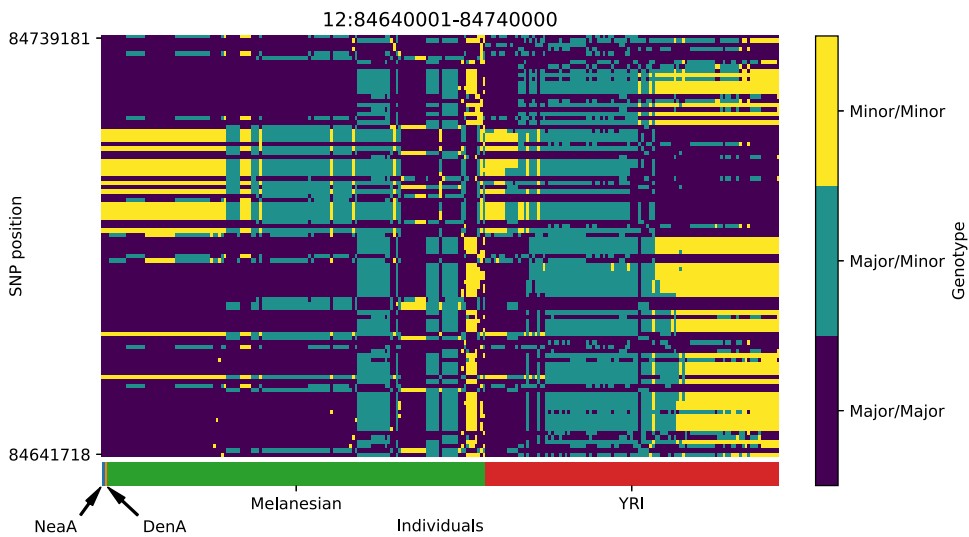

**Appendix 5—figure 12.** Genotype plot for the candidate region chr12:84640001–84740000 in the Denisovan-into-Melanesian AI scan. Dark blue = homozygote major allele, light blue = heterozygote, yellow = homozygote minor allele. Genotypes within populations are sorted left-to-right by similarity to the Denisovan.

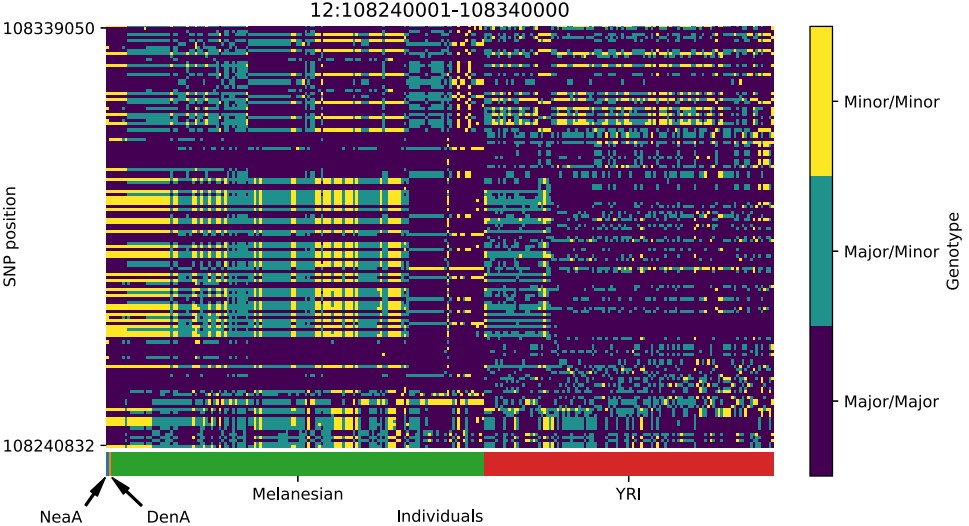

**Appendix 5—figure 13.** Genotype plot for the candidate region chr12:108240001–108340000 in the Denisovan-into-Melanesian AI scan. Dark blue = homozygote major allele, light blue = heterozygote, yellow = homozygote minor allele. Genotypes within populations are sorted left-to-right by similarity to the Denisovan.

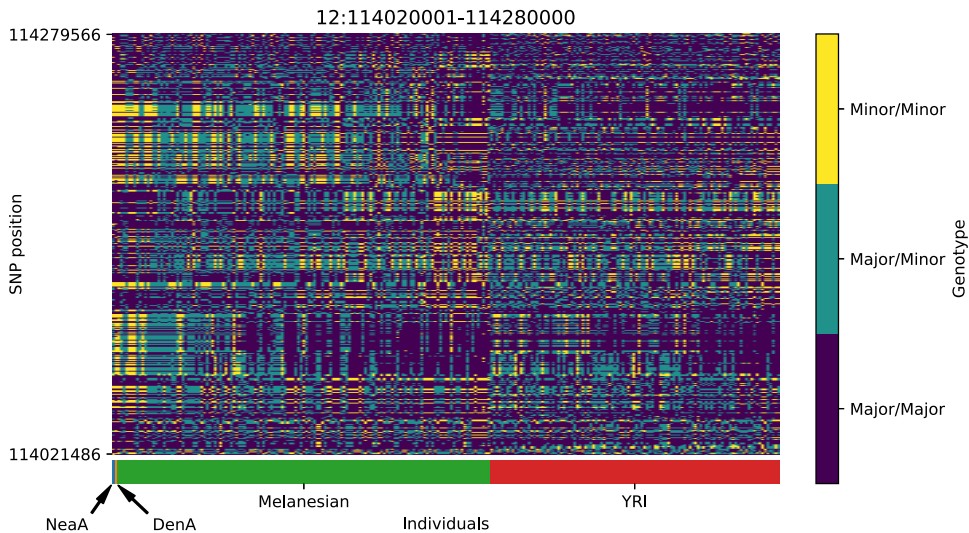

**Appendix 5—figure 14.** Genotype plot for the candidate region chr12:114020001–114280000 in the Denisovan-into-Melanesian AI scan. Dark blue = homozygote major allele, light blue = heterozygote, yellow = homozygote minor allele. Genotypes within populations are sorted left-to-right by similarity to the Denisovan.

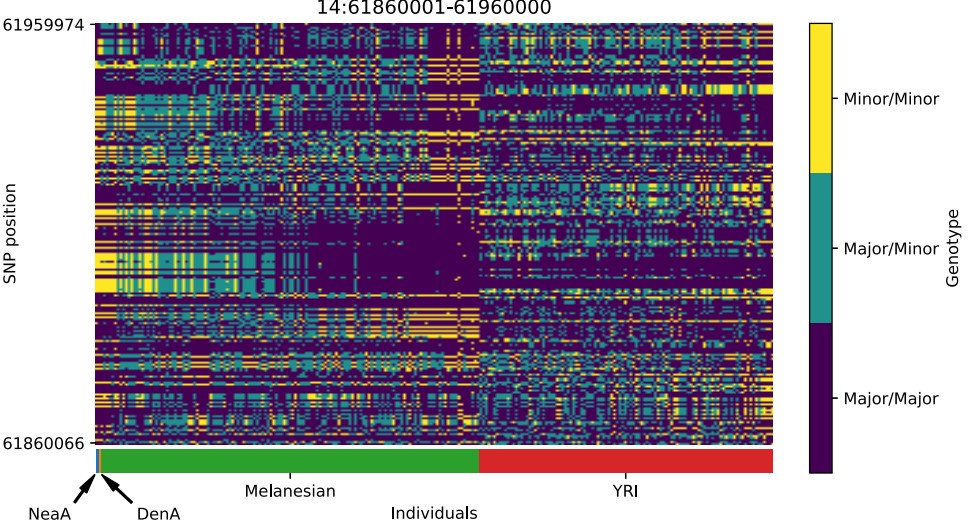

**Appendix 5—figure 15.** Genotype plot for the candidate region chr14:61860001–61960000 in the Denisovan-into-Melanesian AI scan. Dark blue = homozygote major allele, light blue = heterozygote, yellow = homozygote minor allele. Genotypes within populations are sorted left-to-right by similarity to the Denisovan.

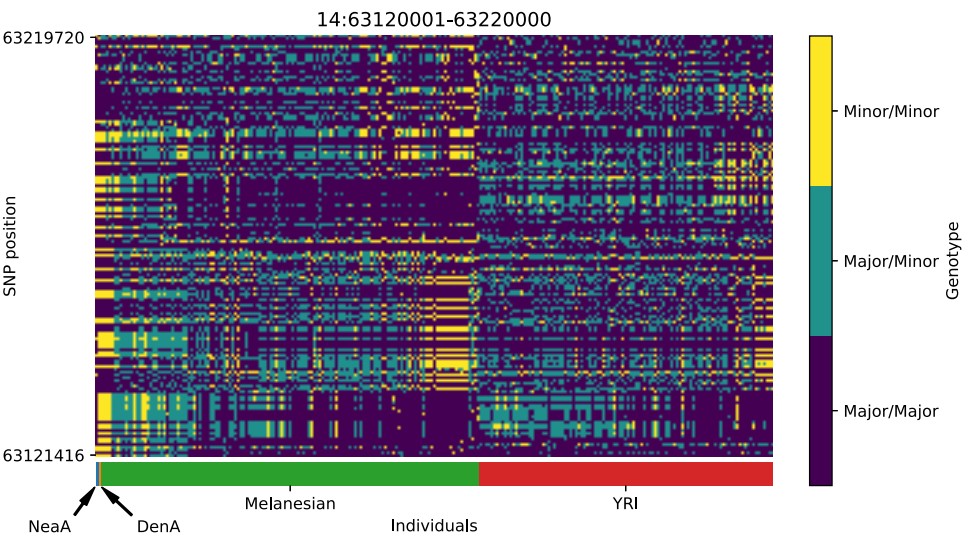

**Appendix 5—figure 16.** Genotype plot for the candidate region chr14:63120001–63220000 in the Denisovan-into-Melanesian AI scan. Dark blue = homozygote major allele, light blue = heterozygote, yellow = homozygote minor allele. Genotypes within populations are sorted left-to-right by similarity to the Denisovan.

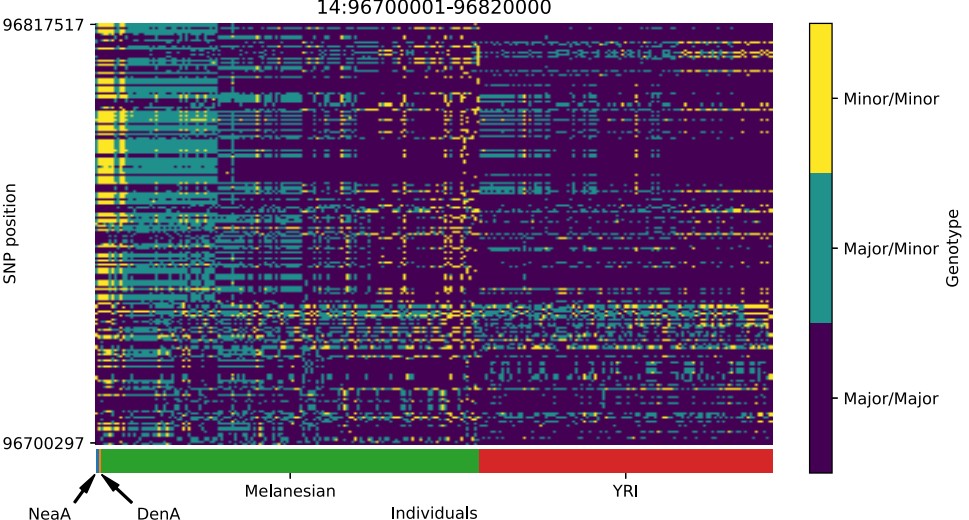

**Appendix 5—figure 17.** Genotype plot for the candidate region chr14:96700001–96820000 in the Denisovan-into-Melanesian AI scan. Dark blue = homozygote major allele, light blue = heterozygote, yellow = homozygote minor allele. Genotypes within populations are sorted left-to-right by similarity to the Denisovan.

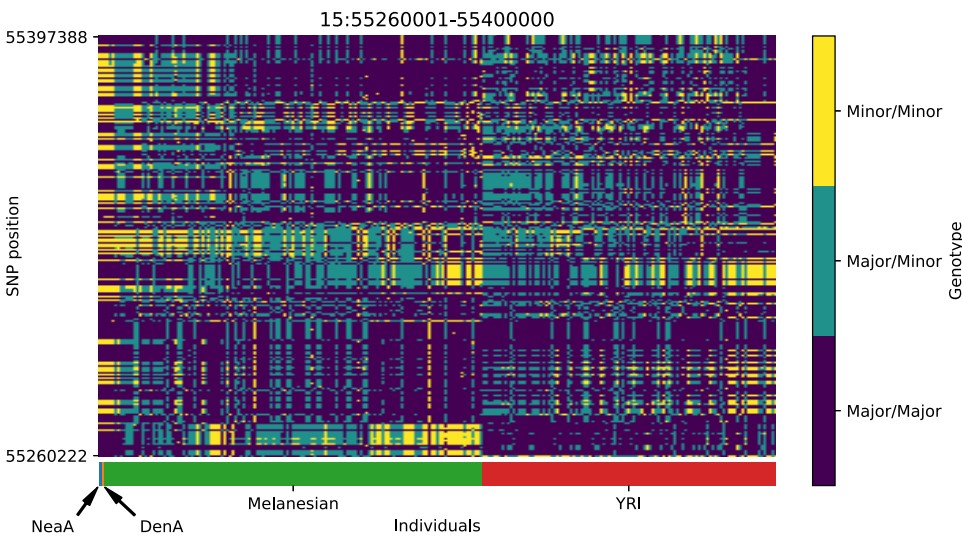

**Appendix 5—figure 18.** Genotype plot for the candidate region chr15:55260001–55400000 in the Denisovan-into-Melanesian AI scan. Dark blue = homozygote major allele, light blue = heterozygote, yellow = homozygote minor allele. Genotypes within populations are sorted left-to-right by similarity to the Denisovan.

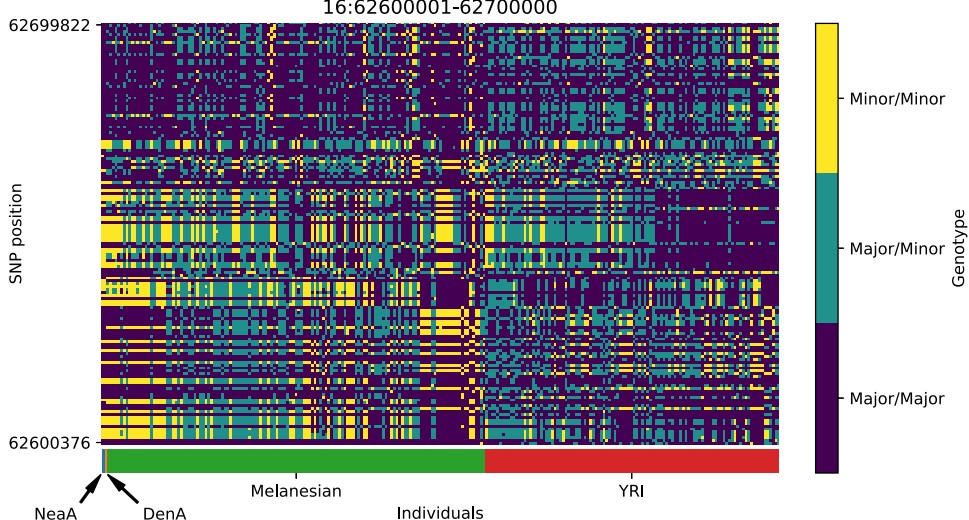

**Appendix 5—figure 19.** Genotype plot for the candidate region chr16:62600001–62700000 in the Denisovan-into-Melanesian AI scan. Dark blue = homozygote major allele, light blue = heterozygote, yellow = homozygote minor allele. Genotypes within populations are sorted left-to-right by similarity to the Denisovan.

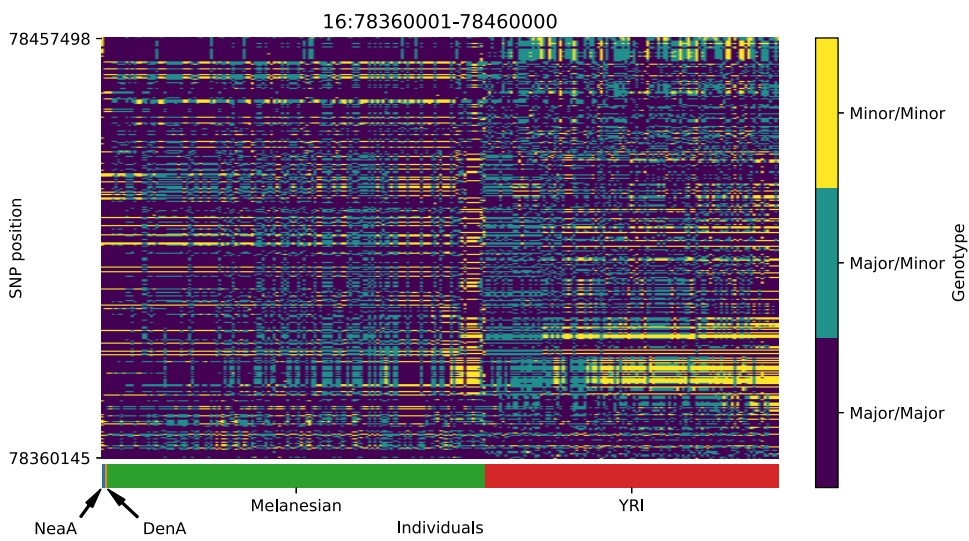

**Appendix 5—figure 20.** Genotype plot for the candidate region chr16:78360001–78460000 in the Denisovan-into-Melanesian AI scan. Dark blue = homozygote major allele, light blue = heterozygote, yellow = homozygote minor allele. Genotypes within populations are sorted left-to-right by similarity to the Denisovan.

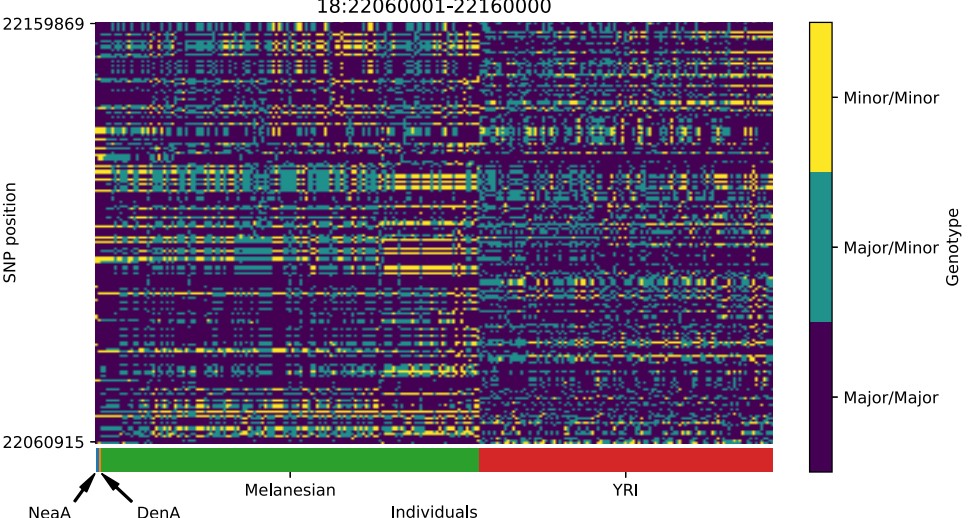

**Appendix 5—figure 21.** Genotype plot for the candidate region chr18:22060001–22160000 in the Denisovan-into-Melanesian AI scan. Dark blue = homozygote major allele, light blue = heterozygote, yellow = homozygote minor allele. Genotypes within populations are sorted left-to-right by similarity to the Denisovan.

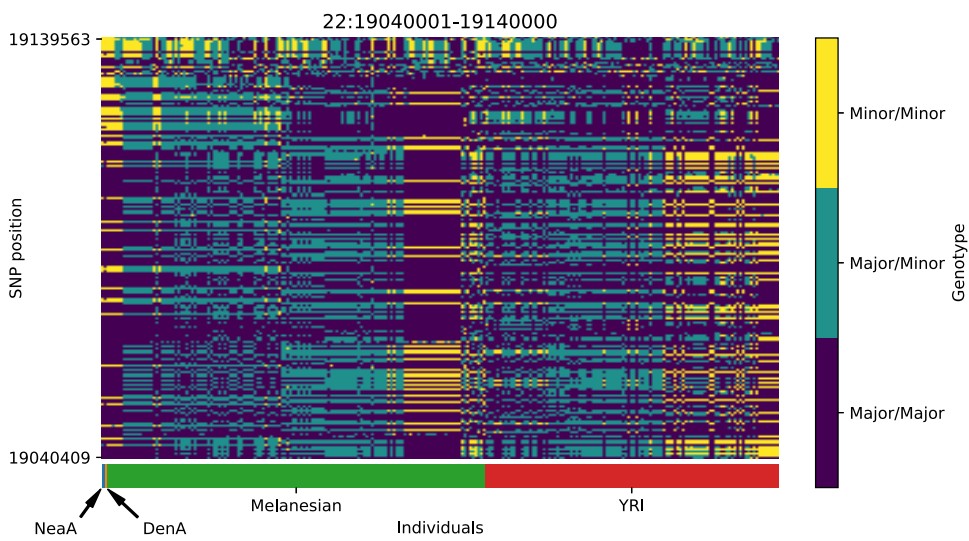

**Appendix 5—figure 22.** Genotype plot for the candidate region chr22:19040001–19140000 in the Denisovan-into-Melanesian AI scan. Dark blue = homozygote major allele, light blue = heterozygote, yellow = homozygote minor allele. Genotypes within populations are sorted left-to-right by similarity to the Denisovan.

