## [Decision Letter]

**Acceptance summary:**

This paper describes a novel approach for detecting adaptive introgression using a deep neural network. The authors demonstrate that their method can accurately detect adaptive introgression under a number of scenarios, and they apply the method to find loci where modern humans may have received beneficial alleles from Neandertals and Denisovans. This application of a convolutional neural network to detect events of adaptive introgression represents an excellent contribution to the field of population and evolutionary genomics.

**Decision letter after peer review:**

Thank you for submitting your article "Detecting adaptive introgression in human evolution using convolutional neural networks" for consideration by *eLife*. Your article has been reviewed by 3 peer reviewers, and the evaluation has been overseen by George Perry as the Senior Editor and Reviewing Editor. The following individual involved in review of your submission has agreed to reveal their identity: Diego Ortega Del Vecchyo (Reviewer #2).

The reviewers have discussed the reviews with one another and the Reviewing Editor has drafted this decision to help you prepare a revised submission.

Summary:

This paper describes a novel approach for detecting adaptive introgression using a deep neural network. The authors demonstrate that their method can accurately detect adaptive introgression under a number of scenarios, and they apply the method to find loci where moderns humans may have received beneficial alleles from Neanderthals and Denisovans. This application of a convolutional neural network to detect events of adaptive introgression represents an excellent contribution to the field of population and evolutionary genomics.

Essential revisions:

I highlight three points of essential revision, and then include the full reviews below which contain additional context on these points as well as a number of other excellent suggestions for your consideration as you revise your paper.

1) Model misidentification. You should test a few more scenarios in which there is demographic model misspecification. For example, you assume that the Yoruba are an 'unadmixed sister population'. Yet recent papers have pointed out that this assumption is potentially incorrect on timescales relevant for your analysis. How does your method perform despite such potential model misidentification? See the individual reviews for several other specific scenarios that would also be ideal examples to test.

2) Comparison with previous methods. The reviewers point out several previous methods to detect adaptive introgression. Please perform a direct comparison of results obtained between your and the previously available approaches.

3) Select a method other than guided propagation for the saliency map, and then evaluate the data along the lines of the detailed suggestions provided by reviewer 3.

*Reviewer #1:*

This manuscript, "Detecting adaptive introgression in human evolution using convolutional neural networks" by Gower et al., proposes a novel approach toward detecting adaptive introgression using a deep neural network. The paper is well written, the authors have taken care to ensure reproducibility and code availability. This method joins a rapidly growing group of deep learning tools for population genetic analysis, and the growing body of evidence that these methods represent an important step forward for the field.

1) I do think that it is essential that the authors compare the power of their method to some existing methods. I know that not all approaches use the exact same information as genomatnn, but I do think that the authors could easily compare power to some of the statistics from Racimo et al., 2017 MBE paper. This would at least offer some context to help readers gauge the level of advance offered by this method.

2) As the authors point out, their accuracy can go down substantially in the case of model misspecification. But they have only examined one scenario--training on Model A and applying to Model B--and this might be unrealistically pessimistic. It would therefore be helpful to see what happens for more modest amounts of misspecification. For example, the authors could sample a few different areas of the parameter space between Model A and Model B and see what happens when the model gets more and more mis-specified. This would give the reader a better feel for how good a demographic model estimate needs to be to use the genomeatnn method in practice.

*Reviewer #2:*

The authors have developed a new method to detect adaptive introgression events using convolutional neural networks (CNN). To use the CNN's, the authors create a training set consisting of 100 kb region simulations where three different scenarios could take place: an adaptive introgression event, a de novo mutation undergoing a sweep without introgression and a scenario without advantageous mutations. The authors use these simulations to train their CNN's to be able to differentiate between an adaptive introgression event happening in the 100 kb region from the latter two scenarios. The methodology presented by the authors includes innovations in the form of codifying the data to run the CNN´s and the CNN architecture used to solve this problem. The authors show that their methodology is able to perform very accurate classifications of regions undergoing adaptive introgression events in realistic human demographic scenarios. Finally, the authors show an application of their method to identify regions undergoing adaptive introgression events from Neanderthals to Europeans, and from Denisovans to Melanesians.

Convolutional neural networks have been recently applied to efficiently solve a variety of problems in population genetics. The application presented by the authors to detect events of adaptive introgression is an excellent and necessary contribution to the field. The manuscript is very well written, and the methods are clearly explained. The method is very robust and I can see that it will be applied to other species as genomic data becomes available. I only have a few minor comments about this manuscript.

The authors assume that YRI is an 'unadmixed sister population'. However, African populations also had introgression with another archaic ghost population (as reported by Ragsdale and Gravel (2019) PLoS Genetics; Durvasula and Sankararaman (2020), Science Advances). Would this have an impact on the detection of introgressed segments from Neanderthals to Europeans or Denisova to Melanesian populations using the method developed by the authors?

*Reviewer #3:*

The authors present in this work a new method based on convolutional neural network to infer adaptive introgression in human population. They trained their network on two types of scenarios with different demographic models. They show that the method works well.

The authors propose interesting features for their network which I think will be of interest of the population genomic and deep learning community.

They also advertise their method as being one of the few to do adaptive introgression inference.

The method is constrained by a demographic model.

The network takes as input a genotype matrix, with sorted and ordered individuals, and m bins in a window of 100kb.

The network is trained as a classification task to say whether the window of interest corresponds to adaptive introgression or not. The output of the sigmoid function (last layer of the CNN), is then used as the probability of AI in the given window.

This probability is then calibrated to take into account the fact that on real world dataset, the categories used in the training phase will likely not match the relative frequency of 100kb region under neutrality, selective sweep or AI.

Finally, the authors applies their method on real dataset and propose new gene as candidate for archaic adaptive introgression.

1. The authors suggest that few methods exist for this task, however, I believe this would be of interest to know how existing methods perform (such as the one from Setter et al., 2020) compared to genomatnn, if possible.

2. The results are good (>95% recall, Figure 2) for scenarios with high selection coefficient and/or early time of onset. The same as Figure 2B but with the precision in addition to recall might be interesting to know whether good precision is found in the same space or in another one (e.g. low selection coeff and late time of onset). Besides, the results seem quite affected by the coefficient of selection, time of onset, and also time of gene flow. Again, having a comparison with other method might help to assess whether the results presented here are good or not, in terms of method. Could this be calibrated as well, or given more weight in the training to improve accuracy for low selection coefficient for instance?

3. Authors do not use a test set. How can we be sure that the author did not "overfit" (unconsciously) the validation dataset after trying different hyperparameters? Showing result on a test set is better practice.

4. The attention analysis should be improved. First, guided backpropagation has been shown to not pass sanity check (Adebayo et al., "Sanity check for saliency maps", NIPS, 2018). It was shown that guided backpropagation highlight pixels that are independent of both the data used for training and the model parameters, and this method should not be used for explanation and interpretation tasks. In this paper, guided backpropagation appears to work merely as an edge detector. I wonder whether this is what is captured by the vertical lines we see in the saliency map that would correspond to clusters made by the rearrangement of the individuals. So I would highly recommend the author to choose another method for the saliency map (that passes sanity check). With the new saliency maps, if there is a signal localized along the SNP dimension, it would be interesting to know whether this actually due to the fact that the beneficial mutation was in the middle of the window, or caused by some other artifact. To do so, the authors could re-simulate scenarios that were easily well classed (e.g. with high selection coefficient and early time of onset), but with the mutation not in the middle.

5. I do no understand why the author propose to study the results with 2 cut-offs of 5 and 25% for the beneficial allele. Also only the 25% cut-off is discussed at the end. So why a cut-off in the first place? Then, why two? And finally, why showing and discussing the results only with 25%? For instance, table 1 and S2 have only 5 regions in common out of 25. How the reader should interpret that?

[Editors' note: further revisions were suggested prior to acceptance, as described below.]

Thank you for resubmitting your work entitled "Detecting adaptive introgression in human evolution using convolutional neural networks" for further consideration by *eLife*. Your revised article has been evaluated by George Perry (Senior Editor) and a Reviewing Editor.

All reviewers expressed appreciation for your hard work and thorough revisions. We have just two relatively small remaining requests before we can consider accepting your paper for publication in *eLife*:

1. You now state in the text: "We note, however, that, as with previous methods, visual inspection of the haplotypes or genotypes of the top candidate regions remains a necessary criterion to accurately assess whether a region may have been under adaptive introgression." We would also like for you to briefly discuss what to do with regions classified as AI are not among the "top candidates". These cases might be harder to validate by visual inspection. Do you think that these cases should be discarded? Or can you share any other insights on how to tell whether these AI regions could be legitimate, and whether users could use this tool to detect the impact of AI across the genome more broadly than just the "top candidates".

2. One previous reviewer comment concerned the absence of a test set (because having only a validation set might lead to overfitting hyperparameters to that set). In your response to reviewer comments you noted that you actually used different simulations and then simulate a new dataset while writing the paper. However, this clarification does not appear in the main text – please revise the manuscript to describe to the reader that the choice of hyperparameters was not made on the validation set mentioned, but instead during the "Preliminary analysis". Otherwise, the reader could think that the network might be biased.

---

## [Author Response]

Essential revisions:I highlight three points of essential revision, and then include the full reviews below which contain additional context on these points as well as a number of other excellent suggestions for your consideration as you revise your paper.

We have addressed all points below.

1) Model misidentification. You should test a few more scenarios in which there is demographic model misspecification. For example, you assume that the Yoruba are an 'unadmixed sister population'. Yet recent papers have pointed out that this assumption is potentially incorrect on timescales relevant for your analysis. How does your method perform despite such potential model misidentification? See the individual reviews for several other specific scenarios that would also be ideal examples to test.

We now include two evaluations of the method with mis-specified demographic models. We retain the existing strongly mis-specified evaluation, training on Demographic Model A1 (Neanderthal/European) and evaluating using model B (Denisovan/Melanesian). In addition, we evaluated the method on a weakly mis-specified model, training on Model A1 and evaluating using model A2 (an extension to model A1 that also includes archaic admixture in Africa as described in Ragsdale and Gravel, 2019). The Results section now reports:

“We then tested robustness to demographic misspecification, by evaluating the CNN trained on Demographic Model A1 against simulations for two other demographic models (Figure 2—figure supplement 2). We considered weak misspecification, where the true demographic history is similar to Demographic Model A1 but also includes archaic admixture within Africa following Ragsdale and Gravel, (2019) (Demographic Model A2; Figure 1—figure supplement 1). This resulted in only a small performance reduction. We also considered strong misspecification, where the true demographic history is Demographic Model B. As there are more Melanesian individuals than European individuals in our simulations (because we aimed to mimic the real number of genomes available in our data analysis below), we down-sampled the Melanesian genomes to match the number of European genomes, so as to perform a fair misspecification comparison. In this case, the performance of the CNN was noticeably worse than that of the summary statistics, but still better than VolcanoFinder. We note that the summary statistics performance decreased also, to match their performance for the correctly-specified assessments on Demographic Model B. Interestingly, we found that the Q95(1%, 100%) statistic was the most robust method for both cases of misspecification.”

To more clearly explain the demographic models, we have added a supplementary figure (Figure 1—figure supplement 1) that shows diagrams of both Demographic Model A1 and A2 together. A diagram of Demographic Model B previously appeared as Figure S1, which we have now removed in favour of a much improved diagram (now Figure 1—figure supplement 2). Writing a demographic model can be error prone (particularly if one must gather the parameters from supplementary material), and so we also provide Demes-format YAML files implementing the three demographic models, available from the genomatnn repository (https://github.com/grahamgower/genomatnn/tree/main/demographic_models). This format is intended to become a de facto standard, used in stdpopsim and elsewhere (see https://popsim-consortium.github.io/demes-spec-docs/).

2) Comparison with previous methods. The reviewers point out several previous methods to detect adaptive introgression. Please perform a direct comparison of results obtained between your and the previously available approaches.

We have now compared the performance of our method to various summary statistics developed by Martin et al. (2015) and Racimo et al., (2017), as well as to the VolcanoFinder method (Setter and Mousset et al., 2020). We note, however, that the latter is meant for cases of a deeply divergent “ghost” adaptive introgression (where data from the source population is absent), and so comparisons with it may not be as appropriate as with the former methods. The results now report:

“We compared the performance of our CNN to VolcanoFinder (Setter et al. 2020), which scans genomes of the recipient population for "volcanoes" of diversity using a coalescent-based model of adaptive introgression (Figure 2—figure supplement 2). However, this method only incorporates information from a single population and performed poorly for the demographic models considered here---in some cases worse than guessing randomly. We also compared our CNN to an outlier-based approach for a range of summary statistics that are sensitive to adaptive introgression (Racimo et al., 2017). Our CNN is closest to a perfect MCC-F1 score for Demographic model A1 and B, closely followed by the Q95(1% ,100%) and then U(1%, 20%, 100%) statistics developed in Racimo et al., (2017).”

To avoid adding a substantial number of figures to the manuscript, and to make fairer and easier comparisons between methods and the different demographic models (correctly and incorrectly specified), we have included a single multi-panel supplementary figure (Figure 2—figure supplement 2) that shows MCC-F1 curves for a range of scenarios. We’ve replaced the TNR/NPV panel in Figure 2C and Figure 2—figure supplement 1C with an MCC-F1 curve and added a section to the results which introduces the MCC-F1:

“While precision, recall, and false positive rate are informative, these each consider only two of the four cells in a confusion matrix (true positives, true negatives, false positives, false negatives), and may produce a distorted view of performance with imbalanced datasets (Chicco, 2017). To obtain a more robust performance assessment, we plotted the Matthews correlation coefficient (MCC; Matthews, 1975) against F1-score (the harmonic mean of precision and recall) for false-positive-rate thresholds from 0 to 100 (Figure 2, Figure 2—figure supplement 1, Figure 2—figure supplement 2), as recently suggested by Cao et al., (2020). MCC produces low scores unless a classifier has good performance in all four confusion matrix cells, and also accounts for class imbalance. In MCC-F1 space, the point (1, 1) indicates perfect predictions, and values of 0.5 for the (unit-normalised) MCC indicate random guessing. These results confirm our earlier findings, that the CNN performance is excellent for Demographic Model A1 when considering either neutral and sweep simulations as the condition negative, and performance decreases slightly when DFE simulations are the negative condition (Figure 2). Furthermore, the CNN performance is not as good for Demographic Model B, but this is unlikely to be caused by using unphased genotypes (Figure 2—figure supplement 1 and Figure 2—figure supplement 2).”

3) Select a method other than guided propagation for the saliency map, and then evaluate the data along the lines of the detailed suggestions provided by reviewer 3.

We have now switched to using the “vanilla” saliency method originally proposed in Simonyan, Vedaldi and Zisserman, (2014), as provided by the tf-keras-vis python package. The results (Figure 3) are very similar, and our interpretation remains the same after these changes. Additionally, we changed the colour scheme for this figure, to more clearly show the bordering pixels --- these indicate a gradient close to zero, but with the previous colour scheme this could have been mistaken for a black border.

Reviewer #1:1) I do think that it is essential that the authors compare the power of their method to some existing methods. I know that not all approaches use the exact same information as genomatnn, but I do think that the authors could easily compare power to some of the statistics from Racimo et al., 2017 MBE paper. This would at least offer some context to help readers gauge the level of advance offered by this method.

We have now compared the performance of our method to various summary statistics developed by Martin et al., (2015) and Racimo et al., (2017), as well as to the VolcanoFinder method (Setter and Mousset et al., 2020). These new results are summarised in the Essential revisions section above.

2) As the authors point out, their accuracy can go down substantially in the case of model misspecification. But they have only examined one scenario--training on Model A and applying to Model B--and this might be unrealistically pessimistic. It would therefore be helpful to see what happens for more modest amounts of misspecification. For example, the authors could sample a few different areas of the parameter space between Model A and Model B and see what happens when the model gets more and more mis-specified. This would give the reader a better feel for how good a demographic model estimate needs to be to use the genomeatnn method in practice.

We have now included a more modest model misspecification scenario, in which the model is trained with Model A1, and then tested in a version of Model A2 in which there is also migration with an archaic African lineage (Ragsdale and Gravel, 2019). These new results are summarised in the Essential revisions section above.

Reviewer #2:The authors have developed a new method to detect adaptive introgression events using convolutional neural networks (CNN). To use the CNN's, the authors create a training set consisting of 100 kb region simulations where three different scenarios could take place: an adaptive introgression event, a de novo mutation undergoing a sweep without introgression and a scenario without advantageous mutations. The authors use these simulations to train their CNN's to be able to differentiate between an adaptive introgression event happening in the 100 kb region from the latter two scenarios. The methodology presented by the authors includes innovations in the form of codifying the data to run the CNN´s and the CNN architecture used to solve this problem. The authors show that their methodology is able to perform very accurate classifications of regions undergoing adaptive introgression events in realistic human demographic scenarios. Finally, the authors show an application of their method to identify regions undergoing adaptive introgression events from Neanderthals to Europeans, and from Denisovans to Melanesians.Convolutional neural networks have been recently applied to efficiently solve a variety of problems in population genetics. The application presented by the authors to detect events of adaptive introgression is an excellent and necessary contribution to the field. The manuscript is very well written, and the methods are clearly explained. The method is very robust and I can see that it will be applied to other species as genomic data becomes available.The authors assume that YRI is an 'unadmixed sister population'. However, African populations also had introgression with another archaic ghost population (as reported by Ragsdale and Gravel (2019) PLoS Genetics; Durvasula and Sankararaman (2020), Science Advances). Would this have an impact on the detection of introgressed segments from Neanderthals to Europeans or Denisova to Melanesian populations using the method developed by the authors?

We now also evaluated our CNN using a model that includes archaic admixture into Yoruba ancestors (Figure 1—figure supplement 1; Ragsdale and Gravel, 2019). The CNN performance is decreased only slightly compared to a correctly specified model (Figure 2—figure supplement 2). These new results are summarised in the Essential revisions section above.

Reviewer #3:The authors present in this work a new method based on convolutional neural network to infer adaptive introgression in human population. They trained their network on two types of scenarios with different demographic models. They show that the method works well.The authors propose interesting features for their network which I think will be of interest of the population genomic and deep learning community.They also advertise their method as being one of the few to do adaptive introgression inference.The method is constrained by a demographic model.The network takes as input a genotype matrix, with sorted and ordered individuals, and m bins in a window of 100kb.The network is trained as a classification task to say whether the window of interest corresponds to adaptive introgression or not. The output of the sigmoid function (last layer of the CNN), is then used as the probability of AI in the given window.This probability is then calibrated to take into account the fact that on real world dataset, the categories used in the training phase will likely not match the relative frequency of 100kb region under neutrality, selective sweep or AI.Finally, the authors applies their method on real dataset and propose new gene as candidate for archaic adaptive introgression.1. The authors suggest that few methods exist for this task, however, I believe this would be of interest to know how existing methods perform (such as the one from Setter et al., 2020) compared to genomatnn, if possible.

We have now made a comparison to VolcanoFinder (Setter and Mousset et al., 2020), as well as several summary statistics (Martin et al., 2015; Racimo et al., 2017). These new results are summarised in the Essential revisions section above. We note that the comparison with VolcanoFinder may not be entirely fair, as Setter et al., themselves suggest their method works poorly for a divergence time between donor-recipient on the scale of Humans and Neanderthals/Denisovans.

2. The results are good (>95% recall, Figure 2) for scenarios with high selection coefficient and/or early time of onset. The same as Figure 2B but with the precision in addition to recall might be interesting to know whether good precision is found in the same space or in another one (e.g. low selection coeff and late time of onset). Besides, the results seem quite affected by the coefficient of selection, time of onset, and also time of gene flow. Again, having a comparison with other method might help to assess whether the results presented here are good or not, in terms of method. Could this be calibrated as well, or given more weight in the training to improve accuracy for low selection coefficient for instance?

Figure 2B shows a heatmap of the true positive rate (aka sensitivity, aka recall: TP/(TP + FN)) across the space of selection coefficient, s, and time of onset of selection, Tsel. The information for this heatmap is easily obtained: the value in any given s/Tsel bin is equivalent to the average Pr{AI} prediction score across AI scenario simulations with s/Tsel values corresponding to the bin. To make an equivalent plot for the precision (TP/(TP + FP)), we would need to also count false positives (FP) corresponding to each s/Tsel bin (a non-AI simulation with the given s/Tsel parameters that was given a Pr[AI]>0.5). But when the condition negative is the DFE or neutral scenario (the majority of our false positives), the simulations cannot be assigned to an s/Tsel bin because there is no positively selected allele and thus no s/Tsel parameters.

The poorer performance of the method in the recent past, and/or with lower selection coefficients, is likely because the haplotypes carrying the beneficial allele had not risen to higher frequency in the recipient population. In the discussion we state:

“for the two putative pulses of Denisovan gene flow (Jacobs et al., 2019), we find our model has greater recall with AI for the more ancient pulse (94% versus 83.6%; Figure 2—figure supplement 1), likely because haplotypes from the older pulse have more time to rise in frequency. Similarly, recall is diminished when the onset of selection is more recent.”

We expect other methods will have difficulty with recent and/or weak selection for the same reason.

3. Authors do not use a test set. How can we be sure that the author did not "overfit" (unconsciously) the validation dataset after trying different hyperparameters? Showing result on a test set is better practice.

We split our data into 90%/10% training/validation sets. The hyperparameters and network architecture were largely tuned on a smaller preliminary set of training/validation simulations that did not vary the selection coefficient or time of onset of selection. After doing simulations that varied the selection coefficient and time of onset of selection, we again split into training/validation sets. Later, when the manuscript was at an advanced draft stage, we discovered our simulations contained a bug, and thus we reran all simulations once again. These latter simulations were used for the training/validation sets reported in the manuscript.

4. The attention analysis should be improved. First, guided backpropagation has been shown to not pass sanity check (Adebayo et al., "Sanity check for saliency maps", NIPS, 2018). It was shown that guided backpropagation highlight pixels that are independent of both the data used for training and the model parameters, and this method should not be used for explanation and interpretation tasks. In this paper, guided backpropagation appears to work merely as an edge detector. I wonder whether this is what is captured by the vertical lines we see in the saliency map that would correspond to clusters made by the rearrangement of the individuals. So I would highly recommend the author to choose another method for the saliency map (that passes sanity check). With the new saliency maps, if there is a signal localized along the SNP dimension, it would be interesting to know whether this actually due to the fact that the beneficial mutation was in the middle of the window, or caused by some other artifact. To do so, the authors could re-simulate scenarios that were easily well classed (e.g. with high selection coefficient and early time of onset), but with the mutation not in the middle.

We thank the reviewer for pointing out the interesting work of Adebayo et al., (2018). We have now switched to using the “vanilla” saliency method originally proposed in Simonyan, Vedaldi and Zisserman, (2014), as provided by the tf-keras-vis python package (as opposed to the keras-vis package we used previously). Note that this package does not include implementations of any “suspect” saliency methods identified by Adebayo et al. The results, and our interpretation, remain the same after these changes.

The vertical bands that appeared in the previous saliency map remain. After much investigation we believe the vertical bands derive from the width of the convolution filter in the first convolution layer, as increasing the filter width does appear to increase the width of the vertical bands in the saliency maps (results not shown).

We have further provided saliency maps for each of the distinct simulation scenarios: neutral, sweep, and AI. Even in the saliency map produced for neutral simulations, it is clear that attention is more concentrated in the middle of the genomic window. This strongly suggests that the trained network has learned to focus here, rather than this being an artefact of edge detection for some classes of input.

5. I do not understand why the author propose to study the results with 2 cut-offs of 5 and 25% for the beneficial allele. Also only the 25% cut-off is discussed at the end. So why a cut-off in the first place ? Then, why two ? And finally, why showing and discussing the results only with 25%? For instance, table 1 and S2 have only 5 regions in common out of 25. How the reader should interpret that?

The motivation for the cut-off was unclear, and we have now fixed that in the text:

“To give the network the best chance of avoiding false positives, we tried two different beneficial-allele frequency cut-offs for training: 5% and 25% (Table 1 and Appendix 1-Table 1). We focus here on describing the results from the 25% condition […]”

The high probability candidates are very similar between the two cut-offs (e.g. see peaks in Figure 4), although when looking at only the top-ranked candidates, we can see differences between the two cut-offs. Naturally, we should expect some variation, because these results are derived from distinct CNNs (the same architecture, but two different training runs), and the training process is stochastic in each case.

[Editors' note: further revisions were suggested prior to acceptance, as described below.]

All reviewers expressed appreciation for your hard work and thorough revisions. We have just two relatively small remaining requests before we can consider accepting your paper for publication in eLife:1. You now state in the text: "We note, however, that, as with previous methods, visual inspection of the haplotypes or genotypes of the top candidate regions remains a necessary criterion to accurately assess whether a region may have been under adaptive introgression." We would also like for you to briefly discuss what to do with regions classified as AI are not among the "top candidates". These cases might be harder to validate by visual inspection. Do you think that these cases should be discarded? Or can you share any other insights on how to tell whether these AI regions could be legitimate, and whether users could use this tool to detect the impact of AI across the genome more broadly than just the "top candidates".

We have added the following paragraph to the discussion below the text quoted above.

“Conversely, there may be regions under AI that are classified as highly probable by the CNN, but that did not appear in our top candidates. Validating a large number of candidates might be difficult, but one could imagine running a differently trained CNN (perhaps one better tailored to distinguish AI from more similar scenarios, like selection on shared ancestral variation) on the subset of the regions that are predicted to be AI using a lenient probability cut-off. One could also use our method more generally, to assess the impact of AI across the genome, by comparing the distribution of probability scores with those of simulation scenarios under different amounts of admixture and selection, though in that case one would need to train the CNN on a wider range of admixture rates and demographic models.”

2. One previous reviewer comment concerned the absence of a test set (because having only a validation set might lead to overfitting hyperparameters to that set). In your response to reviewer comments you noted that you actually used different simulations and then simulate a new dataset while writing the paper. However, this clarification does not appear in the main text – please revise the manuscript to describe to the reader that the choice of hyperparameters was not made on the validation set mentioned, but instead during the "Preliminary analysis". Otherwise, the reader could think that the network might be biased.

We have added the following text to the “CNN model architecture and training” subsection of the Methods:

“The hyperparameters and network architecture were tuned on a smaller preliminary set of simulations that did not vary the selection coefficient or time of onset of selection, so we chose not to split the simulations into a third "test" set when evaluating the models trained on our final simulations.”